# Chemoimmunotherapy Administration Protocol Design for the Treatment of Leukemia through Mathematical Modeling and In Silico Experimentation

**DOI:** 10.3390/pharmaceutics14071396

**Published:** 2022-07-01

**Authors:** Paul A. Valle, Raul Garrido, Yolocuauhtli Salazar, Luis N. Coria, Corina Plata

**Affiliations:** 1Postgraduate Program in Engineering Sciences, BioMath Research Group, Tecnológico Nacional de México/IT Tijuana, Blvd. Alberto Limón Padilla s/n, Tijuana 22454, Mexico; miguel.gonzalezg18@tectijuana.edu.mx (R.G.); luis.coria@tectijuana.edu.mx (L.N.C.); corina.plata@tectijuana.edu.mx (C.P.); 2Postgraduate Program in Engineering, Tecnológico Nacional de México/IT Durango, Blvd. Felipe Pescador 1830 ote., Durango 34080, Mexico; ysalazar@itdurango.edu.mx

**Keywords:** cancer, CAR-T cells, chlorambucil, cytotoxicity, in silico, leukemia, ODEs

## Abstract

Cancer with all its more than 200 variants continues to be a major health problem around the world with nearly 10 million deaths recorded in 2020, and leukemia accounted for more than 300,000 cases according to the Global Cancer Observatory. Although new treatment strategies are currently being developed in several ongoing clinical trials, the high complexity of cancer evolution and its survival mechanisms remain as an open problem that needs to be addressed to further enhanced the application of therapies. In this work, we aim to explore cancer growth, particularly chronic lymphocytic leukemia, under the combined application of CAR-T cells and chlorambucil as a nonlinear dynamical system in the form of first-order Ordinary Differential Equations. Therefore, by means of nonlinear theories, sufficient conditions are established for the eradication of leukemia cells, as well as necessary conditions for the long-term persistence of both CAR-T and cancer cells. Persistence conditions are important in treatment protocol design as these provide a threshold below which the dose will not be enough to produce a cytotoxic effect in the tumour population. In silico experimentations allowed us to design therapy administration protocols to ensure the complete eradication of leukemia cells in the system under study when considering only the infusion of CAR-T cells and for the combined application of chemoimmunotherapy. All results are illustrated through numerical simulations. Further, equations to estimate cytotoxicity of chlorambucil and CAR-T cells to leukemia cancer cells were formulated and thoroughly discussed with a 95% confidence interval for the parameters involved in each formula.

## 1. Introduction

Leukemias are non-solid tumors usually known as cancer of the blood. These hematologic malignancies can be classified as acute or chronic depending on the time-evolution of the disease, and as myelogenous or lymphocytic as they arise from the dysfunctional proliferation of developing leukocytes (white blood cells). Hence, the main classifications are as follows: Acute Myelogenous Leukemia (AML), Chronic Myelogenous Leukemia (CML), Acute Lymphocytic Leukemia (ALL), and Chronic Lymphocytic Leukemia (CLL) [1]. However, regardless of the case, the lifespan of leukemia cancer cells is longer than normal cells and they do not have the ability to fight pathogens as effectively as a normal white blood cell could.

Concerning overall statistics, the Global Cancer Observatory (GLOBOCAN) estimated 474,519 new leukemia cases in 2020 with 311,594 deaths in that year. Here, it should be noted that GLOBOCAN groups leukemias from C91 to C95 (according to the International Classification of Diseases) [2,3]. Nonetheless, lymphocytic (C91) and myelogenous (C92) are the most worrying, where ALL is most common in pediatrics, CLL in the elderly, AML in adults, and, although CML remains the least common, it typically affects older adults and rarely occurs in children.

In the particular case of CLL, since it is the topic of interest in this research, is most frequently diagnosed among people aged 65–74, with a median age at diagnosis of 70 years old, it is rarely seen in people under the age of 40, and is extremely rare in children [4,5]. The diagnosis is established by blood counts with at least 5000 monoclonal lymphocytes per mm^3^ (or µL) for a minimum of three consecutive months; blood smears to look for abnormalities in appearance, number, and shape in these white blood cells; and immunophenotyping of circulating B-lymphocytes, which identify a clonal B-cell population carrying tumor-specific antigens as well as typical B-cell markers [4,6]. CLL is a disease whose progression responds to a clinically heterogeneous picture, making the existence of a curative treatment difficult as the affected individuals have comorbidities due to their advanced age. The latter also implies the appearance of periodic sequences of both response and relapse in many patients. Five-year overall survival rate is estimated in the range of 23–93%, with an overall survival of 2 to more than 20 years. However, survival statistics are based on large groups of people, they cannot be used to predict exactly what will happen to one individual as treatment and response can vary greatly between two patients [5,7,8].

As of today, CLL remains incurable with conventional therapies, and disease progression is inevitable. Concerning chemotherapy treatment regimens for CLL, this may be conducted by several drugs, most notably chlorambucil in the elderly as it is not as toxic as fludarabine, cyclophosphamide, and rituximab. Although chlorambucil remains the treatment of choice for this disease it has shown limited efficacy [8]. Recent clinical advances are aiming for personalized therapy strategies as the new path to follow in cancer treatment. In patients with certain hematologic malignancies such as ALL and CLL, the use of autologous T cells genetically modified to express chimeric antigen receptors (CARs), such as the CD-19, has led to unprecedented clinical responses opening the door to a new era of personalized cancer therapy. Anti-CD19 CAR-T cells may be manufactured from both CD4${}^{+}$ and CD8${}^{+}$ T cell subsets to treat adults with relapsed or refractory CLL [9,10,11].

Despite the excitement around CAR-T cells for the treatment of hematologic malignancies, this therapy has come under criticism for its cost, which in the case of the most recently CAR T-cell therapy approved by the Food and Drug Administration (FDA) is more than USD 450,000 [12]. Nonetheless, clinical trials around the world have been developed to better understand and optimize the application of CAR-T cells [13,14,15,16,17]. On the latter, mathematical and computational modeling coupled with in silico experimentation and nonlinear dynamical systems theories may be a powerful tool in designing personalized chemoimmunotherapy treatment strategies, computer simulations are intended to replace the laborious efficacy testing in real humans and reduce the likelihood of drug failure [18,19,20].

Mathematical models composed of first-order Ordinary Differential Equations (ODEs) have been formulated to describe cancer growth and the effects of therapies such as chemotherapy and immunotherapy, one can see [21,22] for prostate cancer, refs. [23,24] for breast cancer, refs. [25,26,27,28,29] for leukemias, ref. [30] for lung cancer, refs. [31,32,33] for glioma, and many more to explore the overall dynamics between cancer cells and immune-effector cells [34,35,36,37,38,39]. Mathematical models could be tested against the patient clinical information and when additional information about the system becomes available, equations may be reformulated or parameters adjusted accordingly [18].

In recent years, nonlinear system theories such as the Localization of Compact Invariant Sets (LCIS) method, the direct and indirect methods of Lyapunov, and in silico experimentation have been applied to gain some insights in the designing of personalized schedules for the administration of chemotherapy and immunotherapy. However, combined application strategies are still to be explored, which is the main objective of this work. The latter is performed in a mathematical model composed of four first-order ODEs describing the dynamics of alive and dead leukemia cancer cells, CAR-T cells as the immunotherapy treatment, and chlorambucil as the chemotherapy drug. We were able to develop and illustrate diverse protocols of chemoimmunotherapy treatment to control cancer growth and achieve the complete eradication of the leukemia cells in the proposed mathematical model.

The remainder of this paper proceeds as follows. In Section 2, the mathematical model of leukemia and chemoimmunotherapy is formulated, values and units of the parameters are given, and both biological and mathematical assumptions are described. In Section 3, we formulate two equations to estimate cytotoxicity to cancer cells of the chlorambucil chemotherapy drug and the CAR-T cells immunotherapy, as well as how to compute the number of molecules from chlorambucil dose in mg, its concentration in the blood system, and its decay rate by assuming a first-order pharmacokinetics. Further, we present the localizing domain, following with global asymptotic stability conditions that imply the leukemia cells eradication, and persistence of both CAR-T cells and cancer cells. In Section 4, results are illustrated by means of the in silico experimentation, where different cases with treatments and without treatments are explored. In Section 5, both mathematical and numerical simulation results are discussed. Finally, in Section 6, we present the conclusions of our work.

## 2. Mathematical Model: Leukemia and Chemoimmunotherapy

The leukemia and chemoimmunotherapy mathematical model is formulated by considering the dynamics between CLL cancer cells $\left[A\left(t\right)\right]$, dead leukemia cells $\left[A_{d}\left(t\right)\right]$, chemotherapy as chlorambucil $\left[C\left(t\right)\right]$, and immunotherapy in the form of CAR-T cells $\left[T\left(t\right)\right]$ in the blood circulatory system by the following four first-order ODEs:(1)$$\begin{array}{ccc} \dot{A} & = & \rho_{A}A\left(1-\frac{A}{b}\right)-\mu_{A}AA_{d}-\frac{\mu_{AC}AC}{a+C}-\alpha_{T}AT, \end{array}$$(2)$$\begin{array}{ccc} {\dot{A}}_{d} & = & -\mu_{d}A_{d}+\mu_{A}AA_{d}+\frac{\mu_{AC}AC}{a+C}+\alpha_{T}AT, \end{array}$$(3)$$\begin{array}{ccc} \dot{C} & = & -\mu_{C}C-\frac{\mu_{CA}AC}{a+C}-\frac{\mu_{CT}CT}{a+C}+\varphi_{C}, \end{array}$$(4)$$\begin{array}{ccc} \dot{T} & = & \rho_{T}\left(A+T\right)T-\mu_{T}T-\alpha_{T}T^{2}-\frac{\mu_{TC}CT}{a+C}+\varphi_{T}, \end{array}$$
where the time unit is considered as days; $A\left(t\right)$, $A_{d}\left(t\right)$ and $T\left(t\right)$ are given in cells; and $C\left(t\right)$ is measured in molecules. The concentration of cells and molecules in the system is computed when assuming 5 L of blood in a human adult. Further, all solutions with non-negative initial conditions will be located in the non-negative orthant (see Section II.A in [40]):$$R_{+,0}^{4}=\left\{A\left(t\right),A_{d}\left(t\right),T\left(t\right),C\left(t\right)\geq 0\right\}.$$

Interactions among live and dead leukemia cells as well as the chemotherapy units in molecules were introduced by Guzev et al. in their work concerning the experimental validation and in silico experimentation to describe cytotoxicity of three different chemotherapy compounds in leukemia cells [25]. Pérez et al. [27] and León et al. [28] formulated and explored the dynamics of CAR-T cells in the treatment of T-cell and B-cell leukemias, respectively. The latter was further discussed by Valle et al. in [29].

Now, let us discuss interactions in each ODE as follows. The growth of CLL cancer cells $\left[A\left(t\right)\right]$ in the blood circulatory system is defined by the logistic law in the first term of Equation (Equation 1), whereas eradication of these cells is considered by the law of mass action when interacting with dead leukemia cells in the second term; cytotoxicity to leukemia cells from the chemotherapy drug and the CAR-T cells is modeled by the Michaelis–Menten kinetics and the law of mass action in the third and fourth terms, respectively. Further, the maximum tumor carrying capacity is estimated by considering that lymphocytes represent 1.6% of the total amount of cells in a human adult with around $4.5\times 10^{13}$ total cells [41,42]. The evolution of leukemia dead cells $\left[A_{d}\left(t\right)\right]$ is described by Equation (Equation 2), the first term represents dissolution of these cells in the system; remaining terms of this equation indicate accumulation by the eradication of living leukemia cells, the second term describes apoptosis or necrosis as a result of living cells competing for nutrients in the blood circulatory system, while the third and fourth terms are the lysed leukemia cells due to the log-*kill* effect of the chemotherapy and the CAR-T cells immunotherapy, respectively. Pharmacokinetics of the chemotherapy drug $\left[C\left(t\right)\right]$ is given in Equation (Equation 3), where the first term is the decay rate of the drug, this is computed and further explained in Section 3 from its reported biological half-life, term two was introduced by Guzev et al. when considering the total number of molecules attacking each cancer cell; therefore, we incorporated the third term by assuming that the chemotherapy will inevitably have a cytotoxicity effect on the CAR-T cells when combining these two treatments. Constant or periodic administrations of the chemotherapy can be considered by the fourth term. The dynamics of CAR-T cells $\left[T\left(t\right)\right]$ are formulated in Equation (Equation 4), the first term represents stimulation to mitosis, i.e., activation and proliferation, due to encounters with the target antigen in leukemia cancer cells and other CAR-T cells; the second term indicates natural death and/or exhaustion; fratricide or self-inhibition is formulated by the third term; the fourth term considers the cytotoxicity of the chemotherapy in the case that both treatments are present in the system at the same time; and further infusions of the therapy are considered with the fifth term. Description and values of parameters of the leukemia and chemoimmunotherapy mathematical model (Equation 1)–(Equation 4) are shown below in Table 1.

Now, by the cells eradication threshold assumption [20,29,43] that establishes the following: “*If a solution describing the growth of a cell population goes below the value of *1* cell, then it is possible to assume the complete eradication of such population*”, one can formulate the next constraints for all solutions of the leukemia and chemoimmunotherapy system (Equation 1)–(Equation 4):$$\begin{array}{ccc} \mathrm{Live}\;\mathrm{CLL}\;\mathrm{cancer}\;\mathrm{cells} & : & A\left(t\right)=0\;\forall \;A\left(t\right)<1,\\ \mathrm{Dead}\;\mathrm{leukemia}\;\mathrm{cells} & : & A_{d}\left(t\right)=0\;\forall \;A_{d}\left(t\right)<1,\\ \mathrm{Molecules}\;\mathrm{of}\;\mathrm{the}\;\mathrm{chemotherapy}\;\mathrm{drug} & : & C\left(t\right)=0\;\forall \;C\left(t\right)<1,\\ \mathrm{CAR}-T\;\mathrm{cells} & : & T\left(t\right)=0\;\forall \;T\left(t\right)<1. \end{array}$$

Concerning the overall dynamics of CAR-T cells, it should be noted that the parameters regarding their killing efficacy rate $\left(\alpha_{T}\right)$ and the mitosis/activation rate $\left(\rho_{T}\right)$ have the following constraint
(5)$$\alpha_{T}>\rho_{T},$$
providing a mathematical restriction on their proliferation once infused into the system, see [27,28,29]. This condition is directly related to the ultimate bounds of the localizing domain and the leukemia eradication by the immunotherapy treatment. Further, there is a threshold for the numerical value of $\rho_{T}$ that implies either depletion or persistence of the CAR-T cells, and as it is shown below, this value is given by a condition in terms of the death rates of both leukemia $\left(\mu_{A}\right)$ and CAR-T cells $\left(\mu_{T}\right)$, and the dissolution rate of dead leukemia cells $\left(\mu_{d}\right)$ in the blood circulatory system
(6)$$\begin{array}{ccc} \mathrm{Depletion} & : & \rho_{T}<\frac{\mu_{A}\mu_{T}}{\mu_{d}}, \end{array}$$
(7)$$\begin{array}{ccc} \mathrm{Persistence} & : & \rho_{T}>\frac{\mu_{A}\mu_{T}}{\mu_{d}}, \end{array}$$
mathematical background on these conditions will be provided and discussed in Section 3. Regarding the depletion/persistence phenomenon, both circumstances have been reported in in vivo clinical studies. Lee et al. [44] analyzed peripheral blood by flow cytometry and qPCR in 21 patients and reported that CAR-T cells were no longer detected (fewer than $10^{4}$ absolute cells) by day 68 after the last infusion. Porter et al. [7] detected, also by means of flow cytometry and qPCR, persistence of CAR-T cells in the range of 14 to 49 months in the four patients who achieved CR (complete response). However, we have identified that recent studies are leaning to the side of CAR-T cells long-term persistence as new generations of these cells are being developed [11,45,46,47].

## 3. Materials and Methods

### 3.1. Estimating Cytotoxicity of Chemotherapy and CAR-T Cells to Cancer Cells

In this section, nonlinear functions are formulated to estimate the cytotoxicity to cancer cells from different doses of chlorambucil chemotherapy drug and the environment surrounding CAR-T cells immunotherapy. First, let us take the results shown in Table A.1 in Guzev et al. [25]; particularly, chlorambucil concentration (presented in µM), and its corresponding cytotoxicity rate (denoted as $\mu_{AC}$) on CLL cells. As one can see below in Table 2, increasing the drug concentration also increases cytotoxicity. Further, concentration was converted from µM ($10^{-6}$ mol/L) to mg/L, and cytotoxicity rate from h${}^{-1}$ to days ${}^{-1}$ to standardize the time units of the mathematical model under study in this work.

It is evident that chlorambucil cytotoxicity rates are not increasing linearly. Thus, by following the log-*kill* hypothesis, which states that when cancer volume is increasing by a constant fraction of itself every fixed unit of time and it is in the presence of effective anticancer drugs, then it will shrink by a constant fraction of itself [48]. In other words, a therapy dose eradicates a constant proportion of a tumor cell population rather than a constant number of cells. Further, the total amount of eradicated cells will be given by a logarithmic function in base 10. Hence, inspired by the “log” part of the log-*kill* hypothesis we formulate the next nonlinear function to fit the $\mu_{AC}$ values from the column ‘Experimental cytotoxicity in days${}^{-1}$’ in Table 2,
(8)$$\mu_{AC}\left(\varkappa \right)={\left[\log_{10}\left(1+\varepsilon_{1}\varkappa \right)\right]}^{1/\varepsilon_{2}},$$
where ϰ is the chlorambucil concentration in mg/L, and $\mu_{AC}\left(\varkappa \right)$ represents its approximated cytotoxicity rate in days ${}^{-1}$. Experimental and fitted data is illustrated in Figure 1, and the corresponding values are shown in the column ‘Fitted cytotoxicity $\left[\mu_{AC}\left(\varkappa \right)\right]$ in days ${}^{-1}$’ in Table 2 with a coefficient of determination $R^{2}=0.999$. Further, parameters $\varepsilon_{1}$ and $\varepsilon_{2}$ from Equation (Equation 8) were computed with a 95% confidence interval (CI) as it is shown in Table 3.

The mathematical model constructed by Guzev et al. [25] describes the dynamics of the chemotherapy drug in molecules per unit of time. Therefore, let us determine the total number of molecules from its usual dose given in mg as follows [49,50,51]:(9)$$N.Molecules=\frac{\mathrm{Chlorambucil}\;\mathrm{dose}\;\left[\mathrm{mg}\right]\times \mathrm{Avogadro}’s\;\mathrm{number}\;\left[\frac{\mathrm{molecules}}{\mathrm{mol}}\right]}{\mathrm{Chlorambucil}\;\mathrm{average}\;\mathrm{molecular}\;\mathrm{weight}\;\left[\frac{\mathrm{mg}}{\mathrm{mol}}\right]},$$
where
$$\mathrm{Avogadro}’s\;\mathrm{number}=6.02214076\times 10^{23}\;\frac{\mathrm{molecules}}{\mathrm{mol}},$$
and
$$\mathrm{Chlorambucil}\;\mathrm{average}\;\mathrm{molecular}\;\mathrm{weight}=304.212\times 10^{3}\;\frac{\mathrm{mg}}{\mathrm{mol}}.$$

Now, let us explore results shown above by considering the usual protocol of chlorambucil dosing in previously untreated adult patients with CLL. According to the Access Pharmacy Educational Resource [52], off-label dosingis given as indicated below
$$\frac{0.4\;\mathrm{mg}}{\mathrm{kg}\times \mathrm{day}}\;1\;\mathrm{every}\;2\;\mathrm{weeks}\;\mathrm{to}\;a\;\mathrm{maximum}\;\mathrm{of}\;\frac{0.8\;\mathrm{mg}}{\mathrm{kg}\times \mathrm{day}}\;\mathrm{with}\;24\;\mathrm{cycles}.$$

Thus, by assuming a weight of 80 kg (although, in a real-life scenario, the particular weight of each patient can be used) for a CLL adult patient with an average age of ∼70 years [5,53,54], then,
$$\mathrm{Chlorambucil}\;\mathrm{dosing}\;=\;80\;\mathrm{kg}\times \frac{0.4\;\mathrm{mg}}{\mathrm{kg}\times \mathrm{day}}=32\;\frac{\mathrm{mg}}{\mathrm{day}},$$
therefore, the total number of molecules can be computed by Equation (Equation 9) as follows
$$N.Molecules=\frac{32\;\left[\mathrm{mg}\right]\times 6.02214076\times 10^{23}\;\left[\frac{\mathrm{molecules}}{\mathrm{mol}}\right]}{304.212\times 10^{3}\;\left[\frac{\mathrm{mg}}{\mathrm{mol}}\right]},$$
and by simplifying the latter
$$N.Molecules=\left(32\right)\left(1.\;97958685\times 10^{18}\right)\;\mathrm{molecules},$$
the next constant is identified
(10)$$\varsigma =1.\;97958685\times 10^{18},$$
representing the total number of molecules in 1 mg of chlorambucil. At this step, Equation (Equation 9) may be simplified
(11)$$N.Molecules=C_{mg}\times \varsigma ,$$
where $C_{mg}$ is the chlorambucil dose in mg. Hence, for 32 mg we have the following result
$$N.Molecules=32\times \varsigma =6.\;33467792\times 10^{19}\;\mathrm{molecules}.$$

Regarding Equation (Equation 8), it is important to remember that chlorambucil cytotoxicity was investigated by Guzev et al. in micromolar drug concentrations, see column ‘Concentration in mg/L’ in Table 2. Hence, given that an adult has about 5 L of blood [42], in order to properly apply this equation one needs to compute the concentration in the blood circulatory system in mg per liter of blood by dividing as indicated below
$$\mathrm{Chlorambucil}\;\mathrm{concentration},\;\varkappa =\frac{C_{mg}}{5\;L}=\frac{32\;\mathrm{mg}}{5\;L}=6.4\;\frac{\mathrm{mg}}{L},$$
therefore, one can estimate the cytotoxicity of this concentration of chlorambucil to leukemia cells as follows
$$\mu_{AC}\left(\varkappa \right)={\left[\log_{10}\left(1+\left(4.799\times 10^{-2}\right)\left(6.4\right)\right)\right]}^{1/1.693}=0.2806\;{\mathrm{days}}^{-1}.$$

Now, given that the biological half-life $\left(t_{1/2}\right)$ of chlorambucil has been reported as 0.0625 days [50,51], one can determine its deactivation/decay rate (denoted as $\mu_{C}$) from the first-order pharmacokinetics Equation [55]
$$\dot{x}=-\mu_{C}x,$$
with the following solution
$$x\left(t\right)=C_{mg}e^{-\mu_{C}t},$$
for the initial condition $x\left(0\right)=C_{mg}$. Thus, from the latter we have the following
$$\frac{C_{mg}}{2}=C_{mg}e^{-\mu_{C}t_{1/2}},$$
where $C_{mg}/2$ represents 50% of the initial dose at $t_{1/2}$. Therefore, by isolating $\mu_{C}$ we obtain the next deactivation/decay rate for chlorambucil
$$\mu_{C}=\frac{\ln2}{t_{1/2}}=\frac{\ln2}{0.0625\;days}=11.\;090\;{\mathrm{days}}^{-1}.$$

Concerning cytotoxicity of CAR-T cells, Kiesgen et al. [56] made a review study on CAR-T cell-mediated cytotoxicity and concluded that the antitumor efficacy of this ‘living drug’ is influenced by their activation, proliferation and inhibition, as well as their exhaustion. The first three are directly related to their consecutive encounters with cancer and other CAR-T cells, whereas T-cell exhaustion has been reported in many chronic diseases such as human cancer [57]. Furthermore, we were able to estimate cytotoxicity rate values from the Lee et al. [44] phase 1 dose-escalation trial on CAR-T cells for the treatment of leukemia on 21 patients by also considering cancer growth rates. Our results from Table 2 in [29] are summarized below.

Fitted cytotoxicity values of CAR-T cells from column ‘Fitted cytotoxicity $\left[\alpha_{T}\left(\rho_{T},\mu_{T},\rho_{A}\right)\right]$’ in Table 4 were computed with a coefficient of determination $R^{2}=0.995$ by applying the next Equation
(12)$$\alpha_{T}\left(\rho_{T},\mu_{T},\rho_{A}\right)=\gamma_{1}+\frac{\mu_{T}\rho_{T}}{\gamma_{2}+\mu_{T}\rho_{T}}+\frac{\rho_{A}\rho_{T}}{\gamma_{3}+\rho_{A}\rho_{T}},$$
which was formulated by taking into account both the mitosis stimulation rate $\left(\rho_{T}\right)$ and natural death rate of CAR-T cells $\left(\mu_{T}\right)$, as well as the leukemia growth rate $\left(\rho_{A}\right)$. Further discussion on these parameters is provided in Table 1 in Section 2. Parameters $\gamma_{i},i=1,2,3;$ were computed with a 95% CI as it is shown in Table 5.

However, accurately estimating cytotoxicity of CAR-T cells remains as an open problem. Additional clinical data is needed to either validate and/or reformulate Equation (Equation 12). Following this statement, we provide in the Appendix A the necessary code to perform this task in Anaconda 3 with the Jupyter Notebook 6.4.5 (using Python 3.9.7), as all parameters from Equations (Equation 8) and (Equation 12) were fitted by applying the curve_fit function from scipy.optimize [58] with the SciPy Version 1.7.1.

### 3.2. Localization of Compact Invariant Sets

Below, we provide the mathematical background that allows us to determine the localizing domain in the non-negative orthant $R_{+,0}^{4}$ where all compact invariant sets of the leukemia and chemoimmunotherapy system (Equation 1)–(Equation 4) are located. These compact invariant sets can include equilibrium points, periodic orbits, limit cycles and chaotic attractors, among others as illustrated in [59] at Section 3. The so-called General Theorem concerning the LCIS method was formalized by Krishchenko and Starkov in [60] (see Section 2) and it states the following: *Each compact invariant set*
Γ
*of*
$\dot{x}=f\left(x\right)$* is contained in the localizing domain:*$$K\left(h\right)=\left\{h_{\inf}\leq h\left(x\right)\leq h_{\sup}\right\}.$$

From the latter we have that $f\left(x\right)$ is a $C^{\infty }-$ differentiable vector function where $x\in R^{n}$ is the state vector. $h\left(x\right):R^{n}\to R$ is a $C^{\infty }-$ differentiable function called *localizing function*. ${h|}_{S}$ denotes the restriction of $h\left(x\right)$ on a set $S\subset R^{n}$ with $S\left(h\right)=\left\{x\in R^{n}|L_{f}h\left(x\right)=0\right\}$, and $L_{f}h\left(x\right)=\left(\partial h/\partial x\right)f\left(x\right)$ is the Lie derivative of $f\left(x\right)$. Hence, one can define $h_{\inf}=\inf\left\{h\left(x\right)|x\in S\left(h\right)\right\}$ and $h_{\sup}=\sup\left\{h\left(x\right)|x\in S\left(h\right)\right\}$. Furthermore, if all compact invariant sets are contained in the set $K\left(h_{i}\right)$ and in the set $K\left(h_{j}\right)$ then they are contained in $K\left(h_{i}\right)$$\cap K\left(h_{j}\right)$ as well. Nonexistence of compact invariant sets can be considered for a given set $\Lambda \subset R^{n}$ if Λ∩K(h)=∅, then the system $\dot{x}=f\left(x\right)$ has no compact invariant sets located in Λ.

Now, let us explore five localizing functions in order to formulate lower and upper bounds for a localizing domain containing all compact invariant sets of the system (Equation 1)–(Equation 4) in $R_{+,0}^{4}$.

Maximum upper bound for the live leukemia cells population $\left[A\left(t\right)\right]$. In order to determine this bound, the following localizing function is proposed
$$h_{1}=A,$$
and its Lie derivative is defined as follows
$$L_{f}h_{1}=\rho_{A}A\left(1-\frac{A}{b}\right)-\mu_{A}AA_{d}-\frac{\mu_{AC}AC}{a+C}-\alpha_{T}AT,$$
now, one can write set $S\left(h_{1}\right)=\left\{L_{f}h_{1}=0\right\}$ as shown below
$$S\left(h_{1}\right)=\left\{A=b-\frac{b}{\rho_{A}}\left(\mu_{A}A_{d}+\frac{\mu_{AC}C}{a+C}+\alpha_{T}T\right)\right\}\cup \left\{A=0\right\},$$
at this step, negative terms of $S\left(h_{1}\right)$ can be discarded. Hence, non-negative boundaries for all ω− limit sets of $A\left(t\right)$ are given in the next domain
$$K_{A_{1}}=\left\{0\leq A\left(t\right)\leq b\right\}.$$

However, it should be noted that the latter was determined when all the interactions with the other cells and treatments are neglected, thus, the maximum carrying capacity (which is also an equilibrium of $A\left(t\right)$) is located at the upper boundary. Below we will explore another localizing function that considers the effect of the dead leukemia cells in the overall cancer cells population.

Localizing set for the live and the dead leukemia cells populations $\left[A\left(t\right)\;\mathrm{and}\;A_{d}\left(t\right)\right]$. Let us explore the following localizing function
$$h_{2}=A+A_{d},$$
whose Lie derivative is defined as follows
$$L_{f}h_{2}=\rho_{A}A-\frac{\rho_{A}}{b}A^{2}-\mu_{A}AA_{d}-\frac{\mu_{AC}AC}{a+C}-\alpha_{T}AT+\mu_{A}AA_{d}-\mu_{d}A_{d}+\frac{\mu_{AC}AC}{a+C}+\alpha_{T}AT,$$
then set $S\left(h_{2}\right)=\left\{L_{f}h_{2}=0\right\}$ can be formulated and simplified by basic arithmetic operations as follows when considering that $A_{d}=h_{2}-A$
$$S\left(h_{2}\right)=\left\{h_{2}=\left(\frac{\rho_{A}+\mu_{d}}{\mu_{d}}\right)A-\frac{\rho_{A}}{\mu_{d}b}A^{2}\right\},$$
and by completing the square in the right side of the equation we obtain the next result
$$-\frac{\rho_{A}}{\mu_{d}b}A^{2}+\left(\frac{\rho_{A}+\mu_{d}}{\mu_{d}}\right)A=-\frac{\rho_{A}}{\mu_{d}b}{\left(A-\frac{b\left(\rho_{A}+\mu_{d}\right)}{2\rho_{A}}\right)}^{2}+\frac{b{\left(\rho_{A}+\mu_{d}\right)}^{2}}{4\mu_{d}\rho_{A}},$$
therefore,
$$S\left(h_{2}\right)=\left\{h_{2}=\frac{b{\left(\rho_{A}+\mu_{d}\right)}^{2}}{4\mu_{d}\rho_{A}}-\frac{\rho_{A}}{\mu_{d}b}{\left(A-\frac{b\left(\rho_{A}+\mu_{d}\right)}{2\rho_{A}}\right)}^{2}\right\},$$
now, due to $h_{2}=A+A_{d}$ we may conclude the following result
$$K\left(h_{2}\right)=\left\{0\leq A\left(t\right)+A_{d}\left(t\right)\leq \frac{b{\left(\rho_{A}+\mu_{d}\right)}^{2}}{4\mu_{d}\rho_{A}}\right\}.$$

Thus, from the latter, upper bounds for both the live $\left[A\left(t\right)\right]$ and dead $\left[A_{d}\left(t\right)\right]$ leukemia cells populations can be deducted as follows
$$\begin{array}{ccc} K_{A_{2}} & = & \left\{0\leq A\left(t\right)\leq \frac{b{\left(\rho_{A}+\mu_{d}\right)}^{2}}{4\mu_{d}\rho_{A}}\right\},\\ K_{A_{d}} & = & \left\{0\leq A_{d}\left(t\right)\leq A_{d\sup}=\frac{b{\left(\rho_{A}+\mu_{d}\right)}^{2}}{4\mu_{d}\rho_{A}}\right\}. \end{array}$$
Localizing set for the CAR-T cells therapy $\left[T\left(t\right)\right]$. Two localizing functions are analyzed to determine these bounds. First, let us take the next to formulate the lower bound
$$h_{3}=T,$$
whose Lie derivative is defined as follows
$$L_{f}h_{3}=\rho_{T}\left(A+T\right)T-\mu_{T}T-\alpha_{T}T^{2}-\frac{\mu_{TC}CT}{a+C}+\varphi_{T},$$
now, one can write set $S\left(h_{3}\right)=\left\{L_{f}h_{3}=0\right\}$ as indicated below
$$S\left(h_{3}\right)=\left\{-\left(\alpha_{T}-\rho_{T}\right)T^{2}-\left(\mu_{T}+\mu_{TC}\right)T+\varphi_{T}+\frac{a\mu_{TC}T}{a+C}+\rho_{T}AT=0\right\},$$
thus, by condition (Equation 5), i.e., $\alpha_{T}>\rho_{T}$, we can complete the square
$$-\left(\alpha_{T}-\rho_{T}\right)T^{2}-\left(\mu_{T}+\mu_{TC}\right)T=-\left(\alpha_{T}-\rho_{T}\right){\left(T+\frac{\mu_{T}+\mu_{TC}}{2\left(\alpha_{T}-\rho_{T}\right)}\right)}^{2}+\frac{{\left(\mu_{T}+\mu_{TC}\right)}^{2}}{4\left(\alpha_{T}-\rho_{T}\right)},$$
and rewrite $S\left(h_{3}\right)$
$$S\left(h_{3}\right)=\left\{{\left(T+\frac{\mu_{T}+\mu_{TC}}{2\left(\alpha_{T}-\rho_{T}\right)}\right)}^{2}={\left(\frac{\mu_{T}+\mu_{TC}}{2\left(\alpha_{T}-\rho_{T}\right)}\right)}^{2}+\frac{\varphi_{T}}{\alpha_{T}-\rho_{T}}+f\left(A,T\right)\right\},$$
where
$$f_{1}\left(A,C,T\right)=\frac{1}{\alpha_{T}-\rho_{T}}\left(\frac{a\mu_{TC}T}{a+C}+\rho_{T}AT\right),$$
hence, the lower bound is determined by disregarding the non-negative function $f_{1}\left(A,C,T\right)$ and isolating *T*, then, we can conclude on the following lower bound for all solutions of $T\left(t\right)$
$$K_{1}\left(h_{3}\right)=\left\{T\left(t\right)\geq T_{\inf}=\sqrt{{\left(\frac{\mu_{T}+\mu_{TC}}{2\left(\alpha_{T}-\rho_{T}\right)}\right)}^{2}+\frac{\varphi_{T}}{\alpha_{T}-\rho_{T}}}-\frac{\mu_{T}+\mu_{TC}}{2\left(\alpha_{T}-\rho_{T}\right)}\right\},$$
from the latter, it is evident that ${\left.T_{\inf}\right|}_{\varphi_{T}=0}=0$, which is expected in the case where the CAR-T cells therapy is not constantly or periodically applied into the system. Concerning the upper bound, we explore the following localizing function
$$h_{4}=A+T,$$
and by computing its Lie derivative
$$L_{f}h_{4}=\rho_{A}A\left(1-\frac{A}{b}\right)-\mu_{A}AA_{d}-\frac{\mu_{AC}AC}{a+C}-\alpha_{T}AT+\rho_{T}\left(A+T\right)T-\mu_{T}T-\alpha_{T}T^{2}-\frac{\mu_{TC}CT}{a+C}+\varphi_{T},$$
one can define set $S\left(h_{4}\right)=\left\{L_{f}h_{4}=0\right\}$ and write it as follows by considering $T=h_{4}-A$
$$S\left(h_{4}\right)=\left\{\mu_{T}h_{4}=\varphi_{T}-\frac{\rho_{A}}{b}A^{2}+\left(\rho_{A}+\mu_{T}\right)A-\mu_{A}AA_{d}-\frac{\mu_{AC}AC}{a+C}-\left(\alpha_{T}-\rho_{T}\right)\left(AT+T^{2}\right)-\frac{\mu_{TC}CT}{a+C}\right\},$$
now, let us complete the square
$$-\frac{\rho_{A}}{b}A^{2}+\left(\rho_{A}+\mu_{T}\right)A=-\frac{\rho_{A}}{b}{\left(A-\frac{b\left(\rho_{A}+\mu_{T}\right)}{2\rho_{A}}\right)}^{2}+\frac{b{\left(\rho_{A}+\mu_{T}\right)}^{2}}{4\rho_{A}},$$
to rewrite $S\left(h_{4}\right)$
$$S\left(h_{4}\right)=\left\{h_{4}=\frac{\varphi_{T}}{\mu_{T}}+\frac{b{\left(\rho_{A}+\mu_{T}\right)}^{2}}{4\rho_{A}\mu_{T}}-f_{2}\left(A,C,T\right)\right\},$$
where
$$f_{2}\left(A,C,T\right)=\frac{1}{\mu_{T}}\left[\frac{\rho_{A}}{b}{\left(A-\frac{b\left(\rho_{A}+\mu_{T}\right)}{2\rho_{A}}\right)}^{2}+\mu_{A}AA_{d}+\frac{\mu_{AC}AC}{a+C}+\left(\alpha_{T}-\rho_{T}\right)\left(AT+T^{2}\right)+\frac{\mu_{TC}CT}{a+C}\right],$$
therefore, we have the following upper bound for the localizing function $h_{4}$
$$K\left(h_{4}\right)=\left\{A\left(t\right)+T\left(t\right)\leq \frac{\varphi_{T}}{\mu_{T}}+\frac{b{\left(\rho_{A}+\mu_{T}\right)}^{2}}{4\rho_{A}\mu_{T}}\right\}.$$

Now, one can formulate the following lower and upper bounds for all solutions of the live leukemia $\left[A\left(t\right)\right]$ and CAR-T $\left[T\left(t\right)\right]$ cells populations
$$\begin{array}{ccc} K_{T} & = & \left\{T_{\inf}=\sqrt{{\left(\frac{\mu_{T}+\mu_{TC}}{2\left(\alpha_{T}-\rho_{T}\right)}\right)}^{2}+\frac{\varphi_{T}}{\alpha_{T}-\rho_{T}}}-\frac{\left(\mu_{T}+\mu_{TC}\right)}{2\left(\alpha_{T}-\rho_{T}\right)}\leq T\left(t\right)\leq T_{\sup}=\frac{\varphi_{T}}{\mu_{T}}+\frac{b{\left(\rho_{A}+\mu_{T}\right)}^{2}}{4\rho_{A}\mu_{T}}\right\},\\ K_{A_{3}} & = & \left\{0\leq A\left(t\right)\leq A_{\max}=\min\left\{b,\frac{b{\left(\rho_{A}+\mu_{d}\right)}^{2}}{4\mu_{d}\rho_{A}},\frac{\varphi_{T}}{\mu_{T}}+\frac{b{\left(\rho_{A}+\mu_{T}\right)}^{2}}{4\rho_{A}\mu_{T}}\right\}\right\}. \end{array}$$
Localizing set for the chemotherapy drug concentration $\left[C\left(t\right)\right]$. Let us exploit the localizing function denoted by
$$h_{5}=C$$
where its Lie derivative is defined as follows
$$L_{f}h_{5}=-\mu_{C}C-\frac{\mu_{CA}AC}{a+C}-\frac{\mu_{CT}CT}{a+C}+\varphi_{C},$$
now, set $S\left(h_{5}\right)=\left\{L_{f}h_{5}=0\right\}$ can be written as indicated below
$$S\left(h_{5}\right)=\left\{\mu_{C}C=\varphi_{C}-\mu_{CA}A-\mu_{CT}T+\frac{a\mu_{CT}T}{a+C}+\frac{a\mu_{CA}A}{a+C}\right\},$$
hence, by applying the Iterative Theorem
$$S\left(h_{5}\right)\cap K_{A}\cap K_{T}\subset \left\{C\geq \frac{\varphi_{C}}{\mu_{C}}-\frac{\mu_{CA}}{\mu_{C}}A_{\max}-\frac{\mu_{CT}}{\mu_{C}}T_{\sup}\right\},$$
the next lower bound can be determined
$$K_{1}\left(h_{5}\right)=\left\{C\left(t\right)\geq C_{\inf}=\frac{\varphi_{C}-\left(\mu_{CA}A_{\max}+\mu_{CT}T_{\sup}\right)}{\mu_{C}}\right\},$$
from the latter, it is evident that the next condition should be fulfilled
(13)$$\varphi_{C}>\mu_{CA}A_{\max}+\mu_{CT}T_{\sup}.$$
for $C_{\inf}>0$. However, if condition (Equation 13) does not hold, then one should consider $C_{\inf}=0$ as there is no biological sense for negative values for the chemotherapy drug concentration, $C\left(t\right)$. Now, let us take again set $S\left(h_{5}\right)$ as follows
$$S\left(h_{5}\right)=\left\{\mu_{C}C=\varphi_{C}-\frac{\mu_{CA}AC}{a+C}-\frac{\mu_{CT}CT}{a+C}\right\},$$
from which we can determine the next upper bound by disregarding the negative terms as given below
$$K_{2}\left(h_{5}\right)=\left\{C\left(t\right)\leq C_{\sup}=\frac{\varphi_{C}}{\mu_{C}}\right\}.$$

Hence, all solutions $C\left(t\right)$ will be bounded by the following domain
$$K_{C}=\left\{C_{\inf}=\frac{\varphi_{C}-\left(\mu_{CA}A_{\max}+\mu_{CT}T_{\sup}\right)}{\mu_{C}}\leq C\left(t\right)\leq C_{\sup}=\frac{\varphi_{C}}{\mu_{C}}\right\},$$
where $\varphi_{C}\geq 0$ as this term represents the application of the chemotherapy drug into the patient.

Nonexistence conditions and supreme upper bound for the live leukemia cells population $\left[A\left(t\right)\right]$**.** Now, let us apply the Iterative Theorem to the set $S\left(h_{1}\right)$ as follows
$$S\left(h_{1}\right)\cap K_{T}\subset \left\{A\leq b-\frac{\alpha_{T}b}{\rho_{A}}T_{\inf}\right\},$$
from the latter and by assuming (Equation 5) is fulfilled, the next ultimate upper bound can be established
$$K_{A}=\left\{0\leq A\left(t\right)\leq A_{\sup}=b-\frac{\alpha_{T}b}{\rho_{A}}\left(\sqrt{{\left(\frac{\mu_{T}+\mu_{TC}}{2\left(\alpha_{T}-\rho_{T}\right)}\right)}^{2}+\frac{\varphi_{T}}{\alpha_{T}-\rho_{T}}}-\frac{\mu_{T}+\mu_{TC}}{2\left(\alpha_{T}-\rho_{T}\right)}\right)\right\}.$$

Furthermore, one can formulate the following nonexistence condition from $A_{\sup}$ as indicated below
$$A_{\sup}<0,$$
which is solved for the CAR-T cells therapy treatment $\left(\varphi_{T}\right)$
(14)$$\varphi_{T}>\varphi_{CART}=\frac{\rho_{A}}{\alpha_{T}}\left[\frac{\rho_{A}}{\alpha_{T}}\left(\alpha_{T}-\rho_{T}\right)+\mu_{T}+\mu_{TC}\right].$$

Therefore, results shown in this section allow us to conclude the following two statements:

**Theorem** **1.**
*Localizing domain. If condition (Equation 5) holds, then all compact invariant sets of the leukemia and chemoimmunotherapy system (Equation 1)–(Equation 4) are located within the following domain:*

(15)
$$K_{\Gamma }=K_{A}\cap K_{A_{d}}\cap K_{C}\cap K_{T},$$

*with the next lower and upper bounds*

$$\begin{array}{ccc} K_{A} & = & \left\{0\leq A\left(t\right)\leq A_{\sup}\right\},\\ K_{A_{d}} & = & \left\{0\leq A_{d}\left(t\right)\leq A_{d\sup}\right\},\\ K_{C} & = & \left\{C_{\inf}\leq C\left(t\right)\leq C_{\sup}\right\},\\ K_{T} & = & \left\{T_{\inf}\leq T\left(t\right)\leq T_{\sup}\right\}. \end{array}$$



**Corollary** **1.**
*Nonexistence. If condition (Equation 14) fulfills, then there are no compact invariant sets outside the plane A=0 for the leukemia and chemoimmunotherapy system (Equation 1)–(Equation 4).*


### 3.3. Global Asymptotic Stability: Leukemia Cells Eradication

When considering that Equations (Equation 1)–(Equation 4) describe cancer evolution as a nonlinear dynamical system, one can apply Lyapunov’s Direct method (see Chapter 4.1 by Khalil in [61] and Chapter 2 by Hahn in [62]) to investigate the global asymptotic stability of the tumor-free equilibrium point and establish sufficient conditions to ensure the leukemia cells eradication. The latter may be translated into the real-world as immunotherapy doses that could potentially control cancer cells growth. First, let us take the following Lyapunov candidate function $\left(V\right)$
V=A,
and compute its time derivative $\left(\dot{V}\right)$ as follows
$$\dot{V}=\left(\rho_{A}-\frac{\rho_{A}}{b}A-\mu_{A}A_{d}-\frac{\mu_{AC}C}{a+C}-\alpha_{T}T\right)A.$$

Then, in order to fulfill Lyapunov’s asymptotic stability conditions $\left[\dot{V}\left(0\right)=0\;\mathrm{and}\;\dot{V}<0\;\forall \;A>0\right]$, the derivative is evaluated at the localizing domain, i.e.,${\left.\dot{V}\right|}_{K_{\Gamma }}$, and the next upper bound is determined
$$\dot{V}\leq \left(\rho_{A}-\alpha_{T}T_{\inf}\right)A\leq 0,$$
hence, by assuming (Equation 5) holds, we formulate the following condition
$$\rho_{A}-\alpha_{T}\left(\sqrt{{\left(\frac{\mu_{T}+\mu_{TC}}{2\left(\alpha_{T}-\rho_{T}\right)}\right)}^{2}+\frac{\varphi_{T}}{\alpha_{T}-\rho_{T}}}-\frac{\mu_{T}+\mu_{TC}}{2\left(\alpha_{T}-\rho_{T}\right)}\right)<0\;,$$
and solve it for the CAR-T cell therapy parameter as shown below
$$\varphi_{T}>\varphi_{CART}.$$

Therefore, the nonexistence condition (Equation 14) also becomes a sufficient condition for asymptotic stability and the following statement is concluded:

**Theorem** **2.**
*Leukemia cancer cells eradication. If the CAR-T cells therapy dose meets condition (Equation 14), then one can ensure the complete eradication of the leukemia cancer cells population described by the system (Equation 1)–(Equation 4). Thus,*

$$\lim_{t\to \infty }A\left(t\right)=0\;\forall \;A\left(0\right)>0\;\Leftrightarrow \varphi_{T}>\varphi_{CART}.\;$$



Now, one can investigate the leukemia and chemoimmunotherapy system when the live leukemia cancer cells are completely eradicated and both therapies are stopped, i.e., $\left[A\left(t\right),\varphi_{C},\varphi_{T}=0\right]$. Therefore, Equations (Equation 1)–(Equation 4) become as follows
(16)$$\begin{array}{ccc} {\dot{A}}_{d} & = & -\mu_{d}A_{d}, \end{array}$$
(17)$$\begin{array}{ccc} \dot{C} & = & -\mu_{C}C-\frac{\mu_{CT}CT}{a+C}, \end{array}$$
(18)$$\begin{array}{ccc} \dot{T} & = & -\mu_{T}T-\left(\alpha_{T}-\rho_{T}\right)T^{2}-\frac{\mu_{TC}CT}{a+C}, \end{array}$$
where the only biologically meaningful equilibrium point is the leukemia-free state given by
(19)$$\left(A_{0}^{*},A_{d0}^{*},C_{0}^{*},T_{0}^{*}\right)=\left(0,0,0,0\right),$$
hence, the next statement is a direct result from Theorem 2:

**Corollary** **2.***Leukemia-free equilibrium point. If condition (Equation 14) from Theorem 2 is fulfilled and the chemoimmunotherapy treatments are stopped $\left[\varphi_{C},\varphi_{T}=0\right]$* when the leukemia cells are completely eradicated *$\left[A\left(t\right)=0\;\forall \;A\left(t\right)<1\right]$, then the tumor-free equilibrium point (Equation 19) is globally asymptotically stable.*

### 3.4. Persistence: Leukemia and CAR-T Cells

As stated before, long-term persistence of CAR-T cells in cancer patients has been reported in several papers. Thus, this particular phenomenon can be studied as a local asymptotic stability problem by means of Lyapunov’s Indirect method (see Chapter 4.3 by Khalil in [61]) when considering interactions between leukemia and CAR-T cells as a nonlinear system. First, let us simplify the leukemia and chemoimmunotherapy system (Equation 1)–(Equation 4) as follows
(20)$$\begin{array}{ccc} \dot{A} & = & \rho_{A}A\left(1-\frac{A}{b}\right)-\mu_{A}AA_{d}-\alpha_{T}AT, \end{array}$$
(21)$$\begin{array}{ccc} {\dot{A}}_{d} & = & -\mu_{d}A_{d}+\mu_{A}AA_{d}+\alpha_{T}AT, \end{array}$$
(22)$$\begin{array}{ccc} \dot{T} & = & \rho_{T}\left(A+T\right)T-\mu_{T}T-\alpha_{T}T^{2}+\varphi_{T}. \end{array}$$

The latter represents the original system in the short-term as the biological half-life $\left(t_{1/2}\right)$ of chlorambucil is 0.0625 days, which implies that this drug should be depleted in the patient in a short period of time, i.e., $C\left(t\right)=0$. Hence, one can identify the following equilibrium point in order to investigate persistence of CAR-T cells in the mathematical model after the last infusion, i.e., $\varphi_{T}=0$∀$t\in \left[t_{i},\infty \right)$ with $t_{i}>0$,
(23)$$\left(A_{1}^{*},A_{d1}^{*},T_{0}^{*}\right)=\left(\frac{\mu_{d}}{\mu_{A}},\frac{\rho_{A}\left(b\mu_{A}-\mu_{d}\right)}{b\mu_{A}^{2}},0\right),$$
with
(24)$$b\mu_{A}-\mu_{d}>0.$$

Now, as the method requires, the Jacobian matrix $\left[\partial f\left(x\right)/\partial x\right]$ is computed below
(25)$$J=\left[\begin{array}{ccc} \rho_{A}-\frac{2\rho_{A}}{b}A-\mu_{A}A_{d}-\alpha_{T}T & -\mu_{A}A & -\alpha_{T}A\\ \mu_{A}A_{d}+\alpha_{T}T & -\mu_{d}+\mu_{A}A & \alpha_{T}A\\ \rho_{T}T & 0 & \rho_{T}A-2\left(\alpha_{T}-\rho_{T}\right)T-\mu_{T} \end{array}\right],$$
which is evaluated at the Equilibrium point (Equation 23) as follows
$${\left.J\right|}_{\left(A_{1}^{*},A_{d1}^{*},T_{1}^{*}\right)}=\left[\begin{array}{ccc} -\frac{\mu_{d}\rho_{A}}{\mu_{A}b} & -\mu_{d} & -\frac{\alpha_{T}\mu_{d}}{\mu_{A}}\\ \frac{\rho_{A}\left(b\mu_{A}-\mu_{d}\right)}{\mu_{A}b} & 0 & \frac{\alpha_{T}\mu_{d}}{\mu_{A}}\\ 0 & 0 & \frac{\rho_{T}\mu_{d}}{\mu_{A}}-\mu_{T} \end{array}\right],$$
at this step, eigenvalues are computed from the Jacobian determinant $\left[\det\left({\left.J\right|}_{\left(A_{1}^{*},A_{d1}^{*},T_{1}^{*}\right)}-\lambda I\right)=0\right]$, results are shown below
$$\begin{array}{ccc} \lambda_{1} & = & -\frac{\mu_{d}\rho_{A}-\sqrt{\mu_{d}^{2}\rho_{A}^{2}-4\mu_{A}\mu_{d}\rho_{A}b\left(\mu_{A}b-\mu_{d}\right)}}{2\mu_{A}b},\\ \lambda_{2} & = & -\frac{\mu_{d}\rho_{A}+\sqrt{\mu_{d}^{2}\rho_{A}^{2}-4\mu_{A}\mu_{d}\rho_{A}b\left(\mu_{A}b-\mu_{d}\right)}}{2\mu_{A}b},\\ \lambda_{3} & = & -\frac{1}{\mu_{A}}\left(\mu_{A}\mu_{T}-\mu_{d}\rho_{T}\right), \end{array}$$
and by condition (Equation 24) it is evident that *Re*$\lambda_{i}<0$, i=1,2. Therefore, conditions for depletion (Equation 6) and persistence (Equation 7) are derived from $\lambda_{3}$ as follows. If *Re*$\lambda_{3}<0$, then by Theorem 4.7 in [61]: “*The equilibrium is locally asymptotically stable*”, this holds when
$$\rho_{T}<\frac{\mu_{A}\mu_{T}}{\mu_{d}},$$
which biologically implies depletion of the CAR-T cells. Thus, if *Re*$\lambda_{3}>0$, then by Theorem 4.7 in [61]: “*The equilibrium is unstable*”, which holds when
$$\rho_{T}>\frac{\mu_{A}\mu_{T}}{\mu_{d}},$$
and according to Liu and Freedman (see Section 3.1.3 in [63]) this could be biologically interpreted as a “*necessary condition for the cell population to grow*”. Therefore, the following statement is concluded:

**Theorem** **3.**
*CAR-T cells persistence. If conditions (Equation 7) and (Equation 24) hold, then the CAR-T cells immune response to the leukemia cancer cells persists after at least one infusion into the system, i.e.,*

$$\lim_{t\to \infty }T\left(t\right)>0\;when\;A\left(t\right)>0,\;for\;T\left(0\right)>0\;and/or\;\varphi_{T}\left(\tau \right)>0\;with\;\tau \in \left[t_{1},t_{2}\right].$$



Now, following the latter, let us explore leukemia cells persistence under the immunotherapy treatment, i.e., $\varphi_{T}>0$. It is important to note that this implies the CAR-T cells dose will not be enough to control cancer growth. Hence, in this case one should investigate the local stability of the next equilibrium point from the simplified system (Equation 20)–(Equation 23)
(26)$$\left(A_{0}^{*},A_{d0}^{*},T_{1}^{*}\;\right)=\left(0,0,\frac{\sqrt{\mu_{T}^{2}+4\varphi_{T}\left(\alpha_{T}-\rho_{T}\right)}-\mu_{T}}{2\left(\alpha_{T}-\rho_{T}\right)}\right),$$
where $\alpha_{T}>\rho_{T}$ by condition (Equation 5). Thus, the equilibrium (Equation 26) is evaluated at the Jacobian matrix (Equation 25) as indicated below
$${\left.J\right|}_{\left(A_{0}^{*},A_{d0}^{*},T_{0}^{*}\right)}=\left[\begin{array}{ccc} \rho_{A}-\alpha_{T}T_{1}^{*} & 0 & 0\\ \alpha_{T}T_{1}^{*} & -\mu_{d} & 0\\ \rho_{T}T_{1}^{*} & 0 & -2\left(\alpha_{T}-\rho_{T}\right)T_{1}^{*}-\mu_{T} \end{array}\right],$$
as the resulting matrix is lower triangular, all eigenvalues are given by each element of the main diagonal. Therefore,
$$\begin{array}{ccc} \lambda_{4} & = & \rho_{A}-\frac{\alpha_{T}\sqrt{\mu_{T}^{2}+4\varphi_{T}\left(\alpha_{T}-\rho_{T}\right)}-\alpha_{T}\mu_{T}}{2\left(\alpha_{T}-\rho_{T}\right)},\\ \lambda_{5} & = & -\mu_{d},\\ \lambda_{6} & = & -\sqrt{\mu_{T}^{2}+4\varphi_{T}\left(\alpha_{T}-\rho_{T}\right)}, \end{array}$$
and it is evident that $\lambda_{j}<0$, j=5,6. Thus, local asymptotic stability of (Equation 26) follows from the next condition on $\lambda_{4}$
$$\rho_{A}-\frac{\alpha_{T}\sqrt{\mu_{T}^{2}+4\varphi_{T}\left(\alpha_{T}-\rho_{T}\right)}-\alpha_{T}\mu_{T}}{2\left(\alpha_{T}-\rho_{T}\right)}<0,$$
which is solved for the immunotherapy treatment parameter
(27)$$\varphi_{T}>\varphi_{prstnc}=\frac{\rho_{A}}{\alpha_{T}}\left[\frac{\rho_{A}}{\alpha_{T}}\left(\alpha_{T}-\rho_{T}\right)+\mu_{T}\right].$$

In the biological sense, if condition (Equation 27) holds, then the immunotherapy could be able to eradicate a sufficiently small initial tumor population, i.e., the equilibrium point (Equation 26) is locally asymptotically stable. Hence, given the following condition
(28)$$\varphi_{T}<\varphi_{prstnc},$$
the next statement can be concluded regarding the persistence of the leukemia cells population in the system (Equation 1)–(Equation 4):

**Theorem** **4.**
*Leukemia cells persistence. If the CAR-T cells dose meets condition (Equation 28), then the immunotherapy treatment will not be able to eradicate the leukemia cells population. Therefore, cancer persits, i.e.,*

$$\lim_{t\to \infty }A\left(t\right)>0\Leftrightarrow \varphi_{T}<\varphi_{prstnc}.$$



## 4. Results: In Silico Experimentation

In this section, we will explore by means of numerical simulations the overall dynamics of the leukemia and chemoimmunotherapy system (Equation 1)–(Equation 4). It is important to note that our mathematical model aims to describe the evolution of leukemia cancer cells in the blood circulatory system in a hypothetical adult patient when considering the application of immunotherapy in the form of CAR-T cells either alone or combined with the chemotherapy drug chlorambucil. Hence, we formulate four scenarios for the in silico experimentation, i.e., no treatments, CAR-T cells depletion and persistence, and both immunotherapy and chemoimmunotherapy protocols.

Now, it should be noted that the in silico experimentation was performed in Matlab 2022a in a desktop computer with a Ryzen 7 5800X CPU, 64 GB of RAM DDR4 3200, and a 2 TB M.2 Samsung 980 PRO SSD. The system of ODEs (Equation 1)–(Equation 4) was solved by means of Euler’s method, that is
$$x_{i+1}\approx x_{i}+f\left(x\right)\Delta_{t},$$
with a step size $\Delta_{t}=1\times 10^{-7}$ to further reduce the intrinsic error in the system solutions.

### 4.1. No Treatments

First, let us consider the ‘no treatments case’, i.e., $C\left(t\right),T\left(t\right)=0$. Thus, the mathematical model becomes as follows
$$\begin{array}{ccc} \dot{A} & = & \rho_{A}A\left(1-\frac{A}{b}\right)-\mu_{A}AA_{d},\\ {\dot{A}}_{d} & = & -\mu_{d}A_{d}+\mu_{A}AA_{d}, \end{array}$$
where the leukemia cells growth rate $\left(\rho_{A}\right)$ is set to 0.1680. This represents 10% of the rate value of Guzev et al. as this was estimated in vitro in ideal conditions. Further, values of this order of magnitude have been reported in several works concerning mathematical models formulated with in vivo studies of both solid and non-solid tumors [24,26,27,28,34,35,36]. Results are illustrated in Figure 2 with the following initial conditions
$$\begin{array}{ccc} A\left(0\right) & = & 10^{10}\;\mathrm{cells},\\ A_{d}\left(0\right) & = & 0.05\times A\left(0\right)\;\mathrm{cells}, \end{array}$$
which are used through all the in silico experimentation, i.e., Figure 2, Figure 3, Figure 4, Figure 5, Figure 6, Figure 7 and Figure 8. The latter are set to consider a high initial non-solid tumor population and the corresponding 5% of dead leukemia cells proposed by Guzev et al. [25].

### 4.2. CAR-T Cells Depletion and Persistence

Now, regarding depletion and persistence of CAR-T cells, Figure 3 and Figure 4 illustrate, respectively, conditions (Equation 6) and (Equation 7), i.e., ‘CAR-T cells depletion case’ and ‘CAR-T cells persistence case’. For these two scenarios Equations are as follows
$$\begin{array}{ccc} \dot{A} & = & \rho_{A}A\left(1-\frac{A}{b}\right)-\mu_{A}AA_{d}-\alpha_{T}AT,\\ {\dot{A}}_{d} & = & -\mu_{d}A_{d}+\mu_{A}AA_{d}+\alpha_{T}AT,\\ \dot{T} & = & \rho_{T}\left(A+T\right)T-\mu_{T}T-\alpha_{T}T^{2}, \end{array}$$
where $\rho_{T}=2.5714\times 10^{-13}$ for depletion, and $\rho_{T}=2.9409\times 10^{-13}$ for persistence. These values are directly related to the dissolution and dead rates of both leukemia and CAR-T cells as indicated below
$$\begin{array}{ccc} \mathrm{CAR}-T\;\mathrm{cells}\;\mathrm{depletion}\;\mathrm{case} & : & \rho_{T}<\frac{\mu_{A}\mu_{T}}{\mu_{d}},\\ \mathrm{CAR}-T\;\mathrm{cells}\;\mathrm{persistence}\;\mathrm{case} & : & \rho_{T}>\frac{\mu_{A}\mu_{T}}{\mu_{d}}. \end{array}$$

Further, by following the Lee et al. [44] escalation trial the immunotherapy dose was set to $2.4\times 10^{8}$ CAR-T cells as this is the last infusion applied when considering an 80 kg patient, hence
$$T\left(0\right)=2.4\times 10^{8}\;cells,$$
is the initial condition for the in silico experimentation performed in Figure 3 and Figure 4.

### 4.3. CAR-T Cells Treatment Protocols

In the following two cases we explore the leukemia cells eradication when applying the immunotherapy treatment in two different schedules, each one with its corresponding dose. These scenarios are identified as the ‘weekly CAR-T cells application case’ and the ‘fortnight CAR-T cells application case’ as illustrated in Figure 5 and Figure 6, respectively. Numerical simulations are performed by solving the next set of Equations
$$\begin{array}{ccc} \dot{A} & = & \rho_{A}A\left(1-\frac{A}{b}\right)-\mu_{A}AA_{d}-\alpha_{T}AT,\\ {\dot{A}}_{d} & = & -\mu_{d}A_{d}+\mu_{A}AA_{d}+\alpha_{T}AT,\\ \dot{T} & = & \rho_{T}\left(A+T\right)T-\mu_{T}T-\alpha_{T}T^{2}+\varphi_{T}. \end{array}$$

The in silico experimentation allowed us to design two immunotherapy administration protocols with the following characteristics:Weekly CAR-T cells application protocol: Four applications at days 0, 7, 14, and 21; each one with a dose of 202,079,012 cells.Fortnight CAR-T cells application protocol: Four applications at days 0, 14, 28, and 42; each one with a dose of 245,633,284 cells.

Mathematically, the first dose is considered with the initial condition $T\left(0\right)$, whereas the last three consecutive doses were performed with the treatment parameter $\varphi_{T}$ in the form of a delayed pulse train with asymmetrical waves. The experimentation indicates that delaying doses implies an increase in the total number of CAR-T cells that needs to be infused into the patient to achieve cancer eradication, and it should be noted that each immunotherapy dose fulfills the leukemia cancer cells eradication condition (Equation 14) by several orders of magnitude as
$$\varphi_{CART}=\frac{\rho_{A}}{\alpha_{T}}\left[\frac{\rho_{A}}{\alpha_{T}}\left(\alpha_{T}-\rho_{T}\right)+\mu_{T}+\mu_{TC}\right]≃320,252\;\mathrm{CAR}-T\;\mathrm{cells},$$
when considering the next set of parameter values: $\rho_{A}=0.1680$, $\rho_{T}=2.9409\times 10^{-13}$, $\mu_{T}=1/14$, $\alpha_{T}=1.2560\times 10^{-7}$ by Equation (Equation 12), and $\mu_{TC}=0$ as the chemotherapy treatment is not applied in these two cases. Regarding the cytotoxic effect of the therapy, one can see in the upper panel of Figure 5 and Figure 6 that each application produces a 2 to 3log-*kill* of leukemia cells. Nonetheless, numerical simulations also illustrate that cancer cells could begin to grow again if the treatment is stopped.

These two cases allow us to conclude that when immunotherapy is the only treatment applied, then reducing the period between applications yields a better result concerning the number of CAR-T cells needed to control and eradicate the leukemia cells population described by the mathematical model under study in this work. Further, the lower panel in Figure 5 and Figure 6 shows that CAR-T cells population is eventually depleted once leukemia cancer cells have been eradicated after the last application of the therapy. The latter is to be expected as the system becomes as follows
$$\dot{T}=-\mu_{T}T-\left(\alpha_{T}-\rho_{T}\right)T^{2},$$
when $A\left(t\right),A_{d}\left(t\right)=0$, and this equation has only one biologically feasible equilibrium point given by $T^{*}=0$, which is globally asymptotically stable. Hence, long-term persistence of CAR-T cells could be related to the survival of a small tumor population.

### 4.4. Chemoimmunotherapy Treatment Protocols

Now, regarding the combined chemoimmunotherapy treatment strategy, we formulated the next two scenarios: ‘constant chemoimmunotherapy case’ and ‘increasing chemoimmunotherapy case’. As it was stated in Section 3, we considered an 80 kg CLL adult patient with an average age of ∼70 years. In silico experimentations of these two cases were performed with the complete mathematical model (Equation 1)–(Equation 4), i.e.,
$$\begin{array}{ccc} \dot{A} & = & \rho_{A}A\left(1-\frac{A}{b}\right)-\mu_{A}AA_{d}-\frac{\mu_{AC}AC}{a+C}-\alpha_{T}AT,\\ {\dot{A}}_{d} & = & -\mu_{d}A_{d}+\mu_{A}AA_{d}+\frac{\mu_{AC}AC}{a+C}+\alpha_{T}AT,\\ \dot{C} & = & -\mu_{C}C-\frac{\mu_{CA}AC}{a+C}-\frac{\mu_{CT}CT}{a+C}+\varphi_{C},\\ \dot{T} & = & \rho_{T}\left(A+T\right)T-\mu_{T}T-\alpha_{T}T^{2}-\frac{\mu_{TC}CT}{a+C}+\varphi_{T}. \end{array}$$

In order to properly combine the two therapies, we continue with the fortnight CAR-T cells application protocol, i.e., one application every two weeks, and our aim was to eliminate the fourth dose illustrated in Figure 6. First, constant dose applications of the chlorambucil chemotherapy drug were explored. Numerical simulations allowed us to conclude that two chemotherapy administrations after each immunotherapy infusion with a dose of 0.7 mg/kg were necessary to achieve the leukemia cells eradication, as it is shown in Figure 7.

Concerning the increasing dose of chemotherapy, by means of the in silico experimentation we were able to formulate the protocol illustrated in the lower panel of Figure 8, which is as follows
Dose 1 at day 6 with 0.5 $\mathrm{mg}/\mathrm{kg}\to C_{mg}=40$ mg,Dose2atdays10and20with0.6$\;\mathrm{mg}/\mathrm{kg}\to \;C_{mg}=48\;\mathrm{mg},$Dose 3atdays24and34with0.7 $\mathrm{mg}/\mathrm{kg}\to \;C_{mg}=56\;\mathrm{mg},$Dose 4atday38with0.8 $\mathrm{mg}/\mathrm{kg}\to \;C_{mg}=64\;\mathrm{mg}.$

From the latter, one is able to compute the number of molecules of chlorambucil (Equation 11), the concentration in the circulatory system in mg per liter of blood $\left(\varkappa \right)$, and both cytotoxicity to cancer cells (Equation 8) and CAR-T cells as indicated below in Table 6.

Now, the leukemia cells eradication condition (Equation 14) from Theorem 2 changes as the chemotherapy cytotoxicity increases.
$$\begin{array}{ccc} \varphi_{CART} & = & \frac{\rho_{A}}{\alpha_{T}}{\left[\frac{\rho_{A}}{\alpha_{T}}\left(\alpha_{T}-\rho_{T}\right)+\mu_{T}+\mu_{TC}\right]}_{\mu_{TC}\;=\;0.0170}≃342,971\;\mathrm{CAR}-T\;\mathrm{cells},\\ \varphi_{CART} & = & \frac{\rho_{A}}{\alpha_{T}}{\left[\frac{\rho_{A}}{\alpha_{T}}\left(\alpha_{T}-\rho_{T}\right)+\mu_{T}+\mu_{TC}\right]}_{\mu_{TC}\;=\;0.0186}≃345,131\;\mathrm{CAR}-T\;\mathrm{cells},\\ \varphi_{CART} & = & \frac{\rho_{A}}{\alpha_{T}}{\left[\frac{\rho_{A}}{\alpha_{T}}\left(\alpha_{T}-\rho_{T}\right)+\mu_{T}+\mu_{TC}\right]}_{\mu_{TC}\;=\;0.0200}≃347,066\;\mathrm{CAR}-T\;\mathrm{cells},\\ \varphi_{CART} & = & \frac{\rho_{A}}{\alpha_{T}}{\left[\frac{\rho_{A}}{\alpha_{T}}\left(\alpha_{T}-\rho_{T}\right)+\mu_{T}+\mu_{TC}\right]}_{\mu_{TC}\;=\;0.0214}≃348,821\;\mathrm{CAR}-T\;\mathrm{cells}, \end{array}$$
where the other values remain as indicated above, and it is evident that each infusion with a dose of 245,633,284 CAR-T cells continues to fulfill this condition. Thus, one can see in Figure 7 and Figure 8 that the leukemia cancer cells population is eradicated between days 38 and 42. The constant chemoimmunotherapy case required six doses of 56 mg of chlorambucil which amounted to 336 mg of the drug, whereas in the increasing case we needed a total of 312 mg, the upper panel of these two figures illustrates minor differences in both the leukemia cells evolution and the time in which they go below the threshold of complete eradication, i.e., one cancer cell [20,29,43].

The cytotoxicity of the chemotherapy drug to the CAR-T cells was estimated by considering the results reported by de Pillis et al. in [36] as a proportion of the cytotoxicity to cancer cells, i.e., $\mu_{TC}=0.054\mu_{AC}\;$. Hence, as the chlorambucil dose is increased, then the values of $\mu_{AC}$ and $\mu_{TC}$ increase as well, this is shown in columns ‘Cytotoxicity $\mu_{AC}\left(\varkappa \right)$’ and ‘Cytotoxicity $\mu_{TC}$’ of Table 6. These increments over time in the chemotherapy doses and parameter values were incorporated in the in silico experimentation illustrated in Figure 8.

## 5. Discussion

First, concerning the analytical results, our methodology is as follows. The LCIS method was applied to determine ultimate bounds to all solutions for the leukemia and chemoimmunotherapy system (Equation 1)–(Equation 4) as given in the localizing domain $K_{\Gamma }$ (see (Equation 15)). From the latter, nonexistence conditions for the tumor population can be derived. Then, following these results one can establish the sufficient condition (Equation 14) on the immunotherapy treatment to ensure the complete eradication of the leukemia cancer cells by means of Lyapunov’s direct method. This condition was only formulated for the CAR-T cells as there are already well established off-label dosing protocols for chlorambucil administration with which the immunotherapy treatment was intended to be combined. Nonetheless, conditions on both therapies could be explored through this analytical procedure. In addition to the global asymptotic stability conditions of the tumor-free equilibrium point (Equation 19), local stability conditions were calculated with Lyapunov’s indirect method to investigate the long-term persistence of both leukemia and CAR-T cells, as given by (Equation 28) and (Equation 7), respectively.

All conditions derived in this research are given in terms of the system parameters. These are summarized below
$$\begin{array}{ccc} \mathrm{Localizing}\;\mathrm{domain}\;\mathrm{and}\;\mathrm{stability}\;(5) & : & \alpha_{T}>\rho_{T},\\ \mathrm{Depletion}\;\mathrm{of}\;\mathrm{CAR}-T\;\mathrm{cells}\;(6) & : & \rho_{T}<\frac{\mu_{A}\mu_{T}}{\mu_{d}},\\ \mathrm{Persistence}\;\mathrm{of}\;\mathrm{CAR}-T\;\mathrm{cells}\;(7) & : & \rho_{T}>\frac{\mu_{A}\mu_{T}}{\mu_{d}},\\ \mathrm{Eradication}\;\mathrm{of}\;\mathrm{leukemia}\;\mathrm{cells}\;(14) & : & \varphi_{T}>\varphi_{CART}=\frac{\rho_{A}}{\alpha_{T}}\left[\frac{\rho_{A}}{\alpha_{T}}\left(\alpha_{T}-\rho_{T}\right)+\mu_{T}+\mu_{TC}\right],\\ \mathrm{Persistence}\;\mathrm{of}\;\mathrm{leukemia}\;\mathrm{cells}\;(28) & : & \varphi_{T}<\varphi_{prstnc}=\frac{\rho_{A}}{\alpha_{T}}\left[\frac{\rho_{A}}{\alpha_{T}}\left(\alpha_{T}-\rho_{T}\right)+\mu_{T}\right], \end{array}$$
from the latter, the first condition (Equation 5) is directly related to the boundedness, local and global asymptotic stability of the system, and it implies that the killing efficacy rate $\left(\alpha_{T}\right)$ of CAR-T cells should be larger than its activation rate $\left(\rho_{T}\right)$. Concerning depletion or long-term persistence of CAR-T cells we found that this phenomenon is directly proportional to the death rates of leukemia $\left(\mu_{A}\right)$ and CAR-T cells $\left(\mu_{T}\right)$, and inversely proportional to the dissolution rate in the blood circulatory system of dead leukemia cells $\left(\mu_{d}\right)$. At the same time, persistence of CAR-T cells could be associated with the existence of a small population of undetectable cancer cells. Now, the last two constraints, (Equation 14) and (Equation 28), provide sufficient and necessary conditions to ensure the complete eradication of the leukemia cells by the immunotherapy treatment, and a threshold on which the CAR-T cells dose will not be able to control any initial size of the non-solid tumor population.

Regarding the in silico experimentation performed in this work, the leukemia and chemoimmunotherapy system (Equation 1)–(Equation 4) was explored under different scenarios. First, numerical simulations illustrate that in the absence of therapies, leukemia cells massively accumulate in the blood circulatory system reaching values of 50,000 cells/µL as shown in Figure 2, which is expected to be lethal in CLL patients. Depletion and persistence of CAR-T cells after the last infusion into the system was found to be strictly related to their activation rate $\left(\rho_{T}\right)$. Both scenarios are illustrated in Figure 3 and Figure 4, where recent studies demonstrate that long-term persistence of CAR-T cells is to be expected as new generations continue to be developed and explored in in vivo clinical trials. In Figure 5, Figure 6, Figure 7 and Figure 8 we design four administration protocols of immunotherapy and chemoimmunotherapy that completely eradicate the CLL cancer population. Here, it is important to remember that we assume that a cells populations described by an ODEs system are eliminated once the corresponding solution goes below the threshold of 1 cell. The latter is essential to properly apply nonlinear systems theory for modeling the dynamics between cells populations, since any value less than one will not represent any biologically meaningful real-life scenario.

When only the immunotherapy treatment is considered, the in silico experimentation illustrates that reducing the period between applications yields a better overall outcome by improving the in vivo toxicity profile of this so-called living drug. Figure 5 shows that by applying the therapy weekly, fewer cells are needed to eradicate an initial leukemia population of $10^{10}$ cells, whereas by increasing the period between applications, a higher dose of immunotherapy should be infused. Furthermore, there is an instant in time when the leukemia population starts to grow again as there are not enough CAR-T cells to control the tumor population, which is illustrated in Figure 6. Given these results, we decided to explore the chemoimmunotherapy scheme for the ‘fortnight CAR-T cells case’ aiming to stop the regrowth of cancer cells by including the chlorambucil drug between the immunotherapy administrations. Additionally, we are taking advantage of the period between doses to avoid, at least to some extent, the cytotoxicity of the chemotherapy on CAR-T cells. Hence, the in silico experimentation was performed in order to design the combined administration protocols illustrated in Figure 7 and Figure 8. As one can see, we were able to discard the fourth CAR-T cells application by incorporating two chlorambucil intakes after each dose. Both the constant and increasing dosing scenarios demonstrated similar results concerning the time at which complete eradication of leukemia cancer cells was achieved and the total dose of chlorambucil administered to the hypothetical CLL patient.

Another contribution of this work is the formulation of two equations to estimate the cytotoxicity to cancer cells of the chemotherapy drug chlorambucil (Equation 8) and the CAR-T cells immunotherapy (Equation 12), to the best of our knowledge, equations of this form have not been proposed before. Equation (Equation 8) was constructed by fitting a base 10 logarithmic function to the data from the in vitro study of Guzev et al. where cytotoxicity directly depends on the concentration in mg/L of the drug; whereas Equation (Equation 12) provides a minimum value for the cytotoxicity of CAR-T cells that could be enhanced by considering the rates of activation and exhaustion of these cells, as well as the tumor growth rate. However, it is important to note that this equation was fitted to the data of a previous work and further research is needed to either validate or reformulate this equation. Additionally, parameters from Equations (Equation 8) and (Equation 12) were computed with a 95% CI, and they fit the corresponding cytotoxicity data with coefficients of determination $\left(R^{2}\right)$ equal to 0.999 and 0.995, respectively.

Nonetheless, it is important to discuss that even if mathematical models are formulated by considering several aspects of the biological or physiological phenomenon under study, which is also known as mechanistic modeling, they still could be considered as an ideal representation of such phenomenon. Hence, time series data from in vivo clinical studies where the evolution of each cell population could be accurately measured or estimated is needed to validate these models or to better fit the values of the proposed parameters in the system. Following this path, we provided in the Appendix A the necessary code to fit real-life data to nonlinear equations in Python.

## 6. Conclusions

With the continuous advancement of computing power in recent decades, coupled with the increasingly affordable prices, the paradigm of exploring complex biological phenomena such as cancer evolution through mathematical modeling and in silico experimentation has provided interesting and promising results in this field. Particularly, we applied nonlinear system theories and combine them with numerical simulations to explore several scenarios of CLL progression when considering two anticancer therapies: CAR-T cells and chlorambucil. Our methodology allowed us to design treatment protocols for the administration of immunotherapy and chemoimmunotherapy that completely eradicate the leukemia cancer cells population described by our proposed mathematical model.

We expect this research to be useful in the designing of administration protocols for cancer treatment as the in silico experimentation illustrates that mathematical modeling and nonlinear system theories could be applied to obtain insights on tumor evolution and the cytotoxicity of combined anticancer therapies. 

## Figures and Tables

**Figure 1 pharmaceutics-14-01396-f001:**
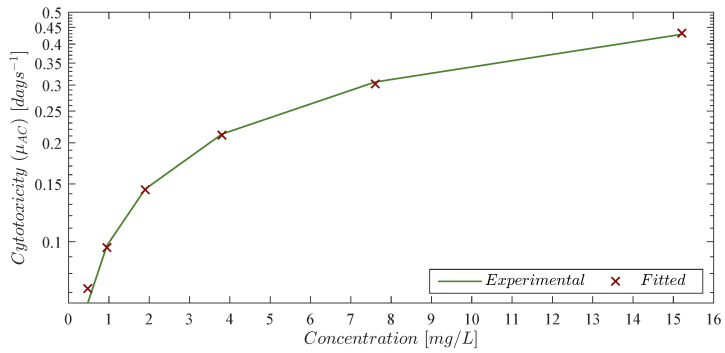
Chlorambucil cytotoxicity rate $\left(\mu_{AC}\right)$ to leukemia cancer cells. The solid green line represents the experimental data obtained from Guzev et al. and the red ‘×’ marker is the estimated value given by Equation (Equation 8).

**Figure 2 pharmaceutics-14-01396-f002:**
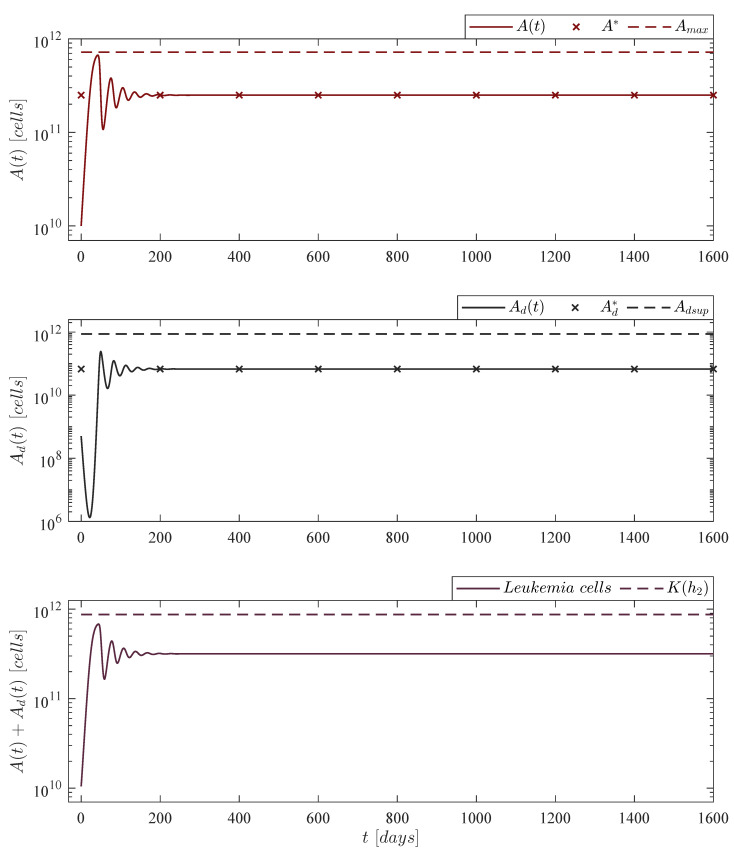
No treatments case. The in silico experimentation illustrates that the absolute count of live leukemia cells $\left[A\left(t\right)\right]$ is of $2.5\times 10^{11}$, which implies 50,000 cancer cells by every µL of blood, while the final count of dead leukemia cells $\left[A_{d}\left(t\right)\right]$ represents ∼26.88% of this value. In the lower panel one can see that the sum of these two cells populations almost reach the maximum carrying capacity but eventually converges to a value below the upper bound given in the localizing set $K\left(h_{2}\right)$. Mathematically, solutions of the live and dead leukemia cells go to the equilibrium point $\left(A^{*},A_{d}^{*}\right)=\left(\mu_{d}/\mu_{A},\rho_{A}\left(b\mu_{A}-\mu_{d}\right)/\left(b\mu_{A}^{2}\right)\right)$. For this case, parameter $\rho_{A}$ is set to 0.1680.

**Figure 3 pharmaceutics-14-01396-f003:**
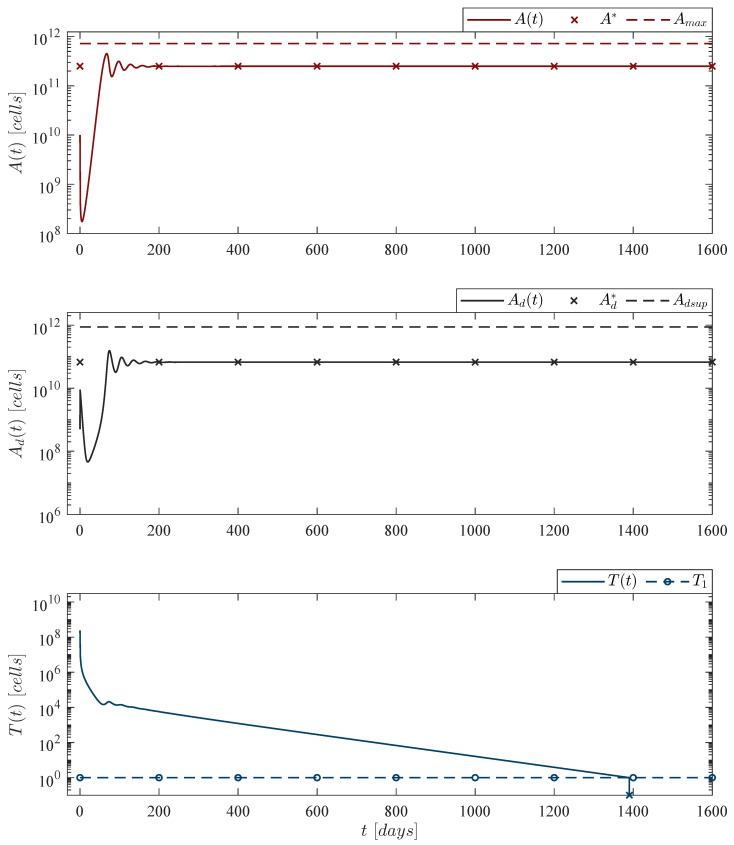
CAR-T cells depletion case. Numerical simulations allow us to illustrate that this phenomenon arises when the mitosis/activation rate $\left(\rho_{T}\right)$ of CAR-T cells is below a threshold directly related to the death rates of both leukemia $\left(\mu_{A}\right)$ and CAR-T cells $\left(\mu_{T}\right)$, and the dissolution rate of dead leukemia cells $\left(\mu_{d}\right)$ as given by condition (Equation 6). The upper panel shows that there is almost a 2log-*kill* of leukemia cells due to the treatment. However, as CAR-T cells are eventually depleted, the leukemia concentration ends as the ‘no treatments case’, i.e., 50,000 cells/µL. For this case, parameter values are set as follows: $\rho_{A}=0.1680$, $\rho_{T}=2.5714\times 10^{-13}$, $\mu_{T}=1/14$, and $\alpha_{T}=1.2560\times 10^{-7}$ by Equation (Equation 12).

**Figure 4 pharmaceutics-14-01396-f004:**
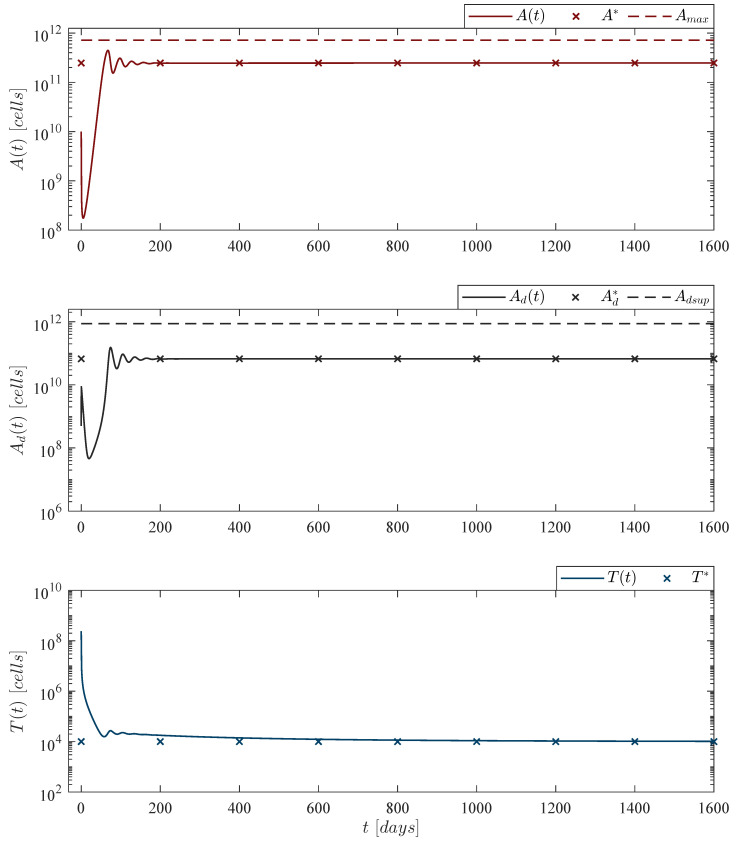
CAR-T cells persistence case. The in silico experimentation illustrates that by fulfilling condition (Equation 7) CAR-T cells can persist after a single application with an absolute count of 10,001 cells, just above the threshold of clinical detectability reported in [44]. Nonetheless, in our mathematical model persistence of CAR-T cells is linked to the survival of leukemia cells. Solutions of live leukemia cells $\left[A\left(t\right)\right]$, dead leukemia cells $\left[A_{d}\left(t\right)\right]$ and CAR-T cells $\left[T\left(t\right)\right]$ go to the equilibrium point $\left(A^{*},A_{d}^{*},T^{*}\right)≃\left(2.4716\times 10^{11},6.6835\times 10^{10},1.0001\times 10^{4}\right)$. The analytical analysis shows that the numerical value of $\rho_{T}$ is directly related to this result. For this case, parameter values are set as follows: $\rho_{A}=0.1680$, $\rho_{T}=2.9409\times 10^{-13}$, $\mu_{T}=1/14$, and $\alpha_{T}=1.2560\times 10^{-7}$ by Equation (Equation 12).

**Figure 5 pharmaceutics-14-01396-f005:**
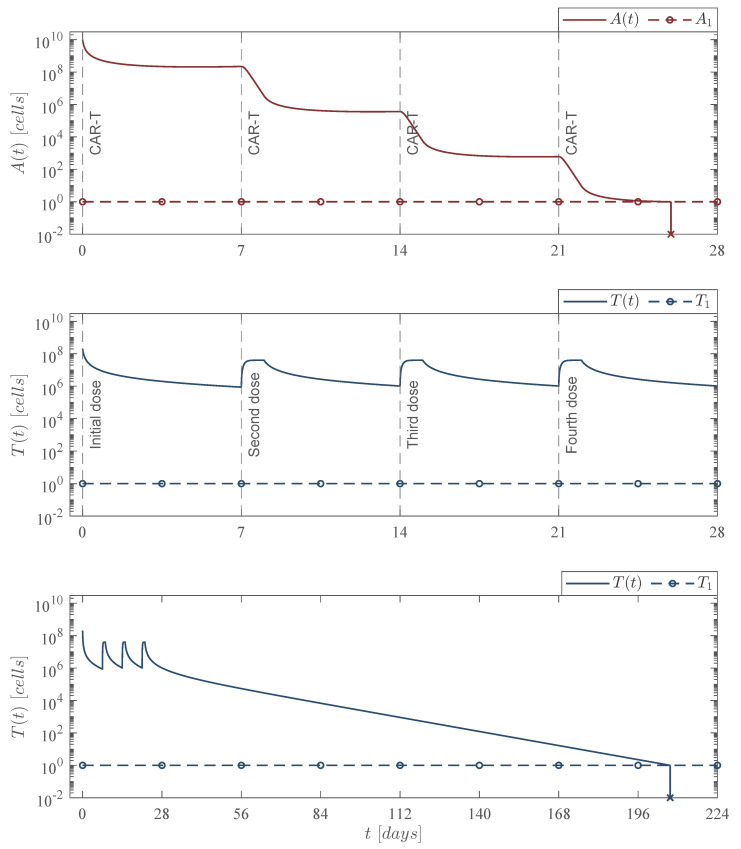
Weekly CAR-T cells case. Four doses of 202,079,012 CAR-T cells were applied at days 0, 7, 14, and 21, for a total of 808,316,048 CAR-T cells infused into the system to achieve leukemia cancer cells eradication $\left[A\left(t\right)<1\right]$ between days 25 and 26 $\left[t=25.96\right]$. The lower panel illustrates that CAR-T cells are completely depleted $\left[T\left(t\right)<1\right]$ by day 208 $\left[t=207.32\right]$. For this case, parameter values are set as follows: $\rho_{A}=0.1680$, $\rho_{T}=2.9409\times 10^{-13}$, $\mu_{T}=1/14$, and $\alpha_{T}=1.2560\times 10^{-7}$ by Equation (Equation 12).

**Figure 6 pharmaceutics-14-01396-f006:**
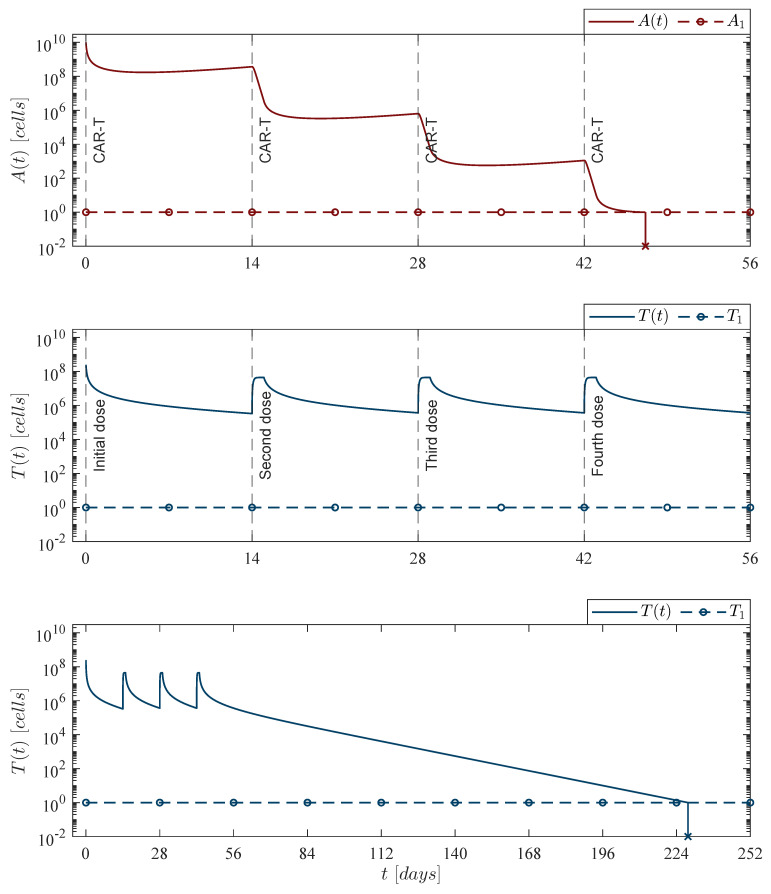
Fortnight CAR-T cells case. Four doses of 245,633,284 CAR-T cells were applied at days 0, 14, 28, and 42, for a total of 982,533,136 CAR-T cells infused into the system to achieve leukemia cancer cells eradication $\left[A\left(t\right)<1\right]$ between days 47 and 48 $\left[t=47.16\right]$. The lower panel illustrates that CAR-T cells are completely depleted $\left[T\left(t\right)<1\right]$ by day 229 $\left[t=228.34\right]$. For this case, parameter values are set as follows: $\rho_{A}=0.1680$, $\rho_{T}=2.9409\times 10^{-13}$, $\mu_{T}=1/14$, and $\alpha_{T}=1.2560\times 10^{-7}$ by Equation (Equation 12).

**Figure 7 pharmaceutics-14-01396-f007:**
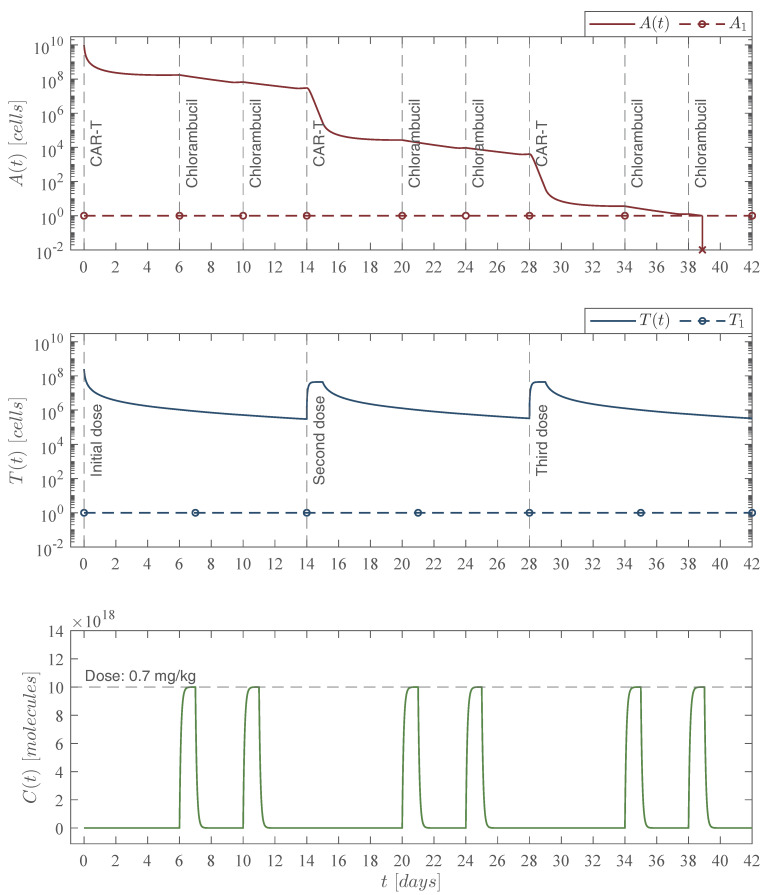
Constant chemoimmunotherapy case. In this scenario we incorporated consecutive administrations of the chlorambucil chemotherapy drug with a constant dose. Numerical simulations allowed us to determine both the amount dose and the day at which the treatment should be applied to achieve leukemia cells eradication. The chemoimmunotherapy protocol was established as follows for a hypothetical 80 kg leukemia patient: three immunotherapy infusions of 245,633,284 CAR-T cells at days 0, 14, and 28 as illustrated in the middle panel; six administrations of the chlorambucil chemotherapy drug with a dose of 56 mg $\left(0.7\;\mathrm{mg}/\mathrm{kg}\right)$ at days 6, 10, 20, 24, 34, and 38. The combined therapy is illustrated in the upper panel with the leukemia cells evolution $\left[A\left(t\right)\right]$ which go below the threshold of complete eradication by day 39 $\left[t=38.87\right]$ for an initial non-solid tumor population of $10^{10}$ cancer cells. Total doses of the chemoimmunotherapy treatment are as follows: 336 mg of chlorambucil, and 736,899,852 CAR-T cells. For this case, parameter values are set as follows: $\rho_{A}=0.1680$, $\rho_{T}=2.9409\times 10^{-13}$, $\mu_{T}=1/14$, $\alpha_{T}=1.2560\times 10^{-7}$ by Equation (Equation 12), $\mu_{AC}=0.3712$, and $\mu_{TC}=0.0200$.

**Figure 8 pharmaceutics-14-01396-f008:**
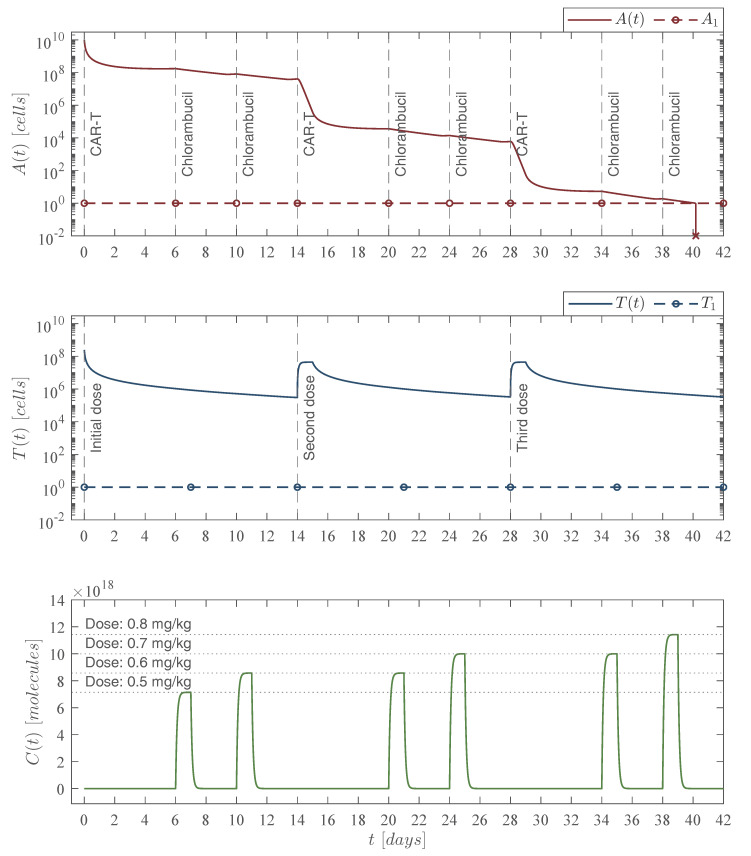
Increasing chemoimmunotherapy case. The in silico experimentation allowed us to determine the increments of the chlorambucil dosing to achieve leukemia cells eradication as follows: 40 mg at day 6, 48 mg at days 10 and 20, 56 mg at days 24 and 34, and 64 mg at day 38 as illustrated in the lower panel. The immunotherapy infusions remain as 245,633,284 CAR-T cells at days 0, 14, and 28 as shown in the middle panel. The combined therapy strategy is illustrated in the upper panel with the leukemia cells evolution $\left[A\left(t\right)\right]$ which go below the threshold of complete eradication by day 41 $\left[t=40.18\right]$ for an initial non-solid tumor population of $10^{10}$ cancer cells. Total doses of the chemoimmunotherapy treatment are as follows: 336 mg of chlorambucil, and 736,899,852 CAR-T cells. It should be noted that the final CAR-T cells count is lower than the weekly and fortnight CAR-T cells cases of Figure 5 and Figure 6, respectively. For this case, parameter values are set as follows: $\rho_{A}=0.1680$, $\rho_{T}=2.9409\times 10^{-13}$, $\mu_{T}=1/14$, and $\alpha_{T}=1.2560\times 10^{-7}$ by Equation (Equation 12). For the values of $\mu_{AC}$ and $\mu_{TC}$ see columns ‘Cytotoxicity $\mu_{AC}\left(\varkappa \right)$’ and ‘Cytotoxicity $\mu_{TC}$’ of Table 6.

**Table 1 pharmaceutics-14-01396-t001:** Parameter information for the leukemia and chemoimmunotherapy mathematical model.

Parameter	Description	Values	Units
$\rho_{A}$	Leukemia cells growth rate	0<r≤1.68	days ${}^{-1}$
*b*	Maximum leukemia cells carrying capacity	$7.2\times 10^{11}$	cells
$\mu_{A}$	Death rate of leukemia cells due to necrosis	$1.63199\times 10^{-12}$	${\left(\mathrm{cells}\times \mathrm{days}\right)}^{-1}$
*a*	Chemotherapy dose that produces 50% maximum effect	$1\times 10^{7}$	molecules
$\mu_{AC}$	Chemotherapy cytotoxicity rate on cancer cells	See Equation (Equation 8)	days ${}^{-1}$
$\alpha_{T}$	Killing efficacy rate (cytotoxicity) of CAR-T cells	See Equation (Equation 12)	${\left(\mathrm{cells}\times \mathrm{days}\right)}^{-1}$
$\mu_{d}$	Dissolution rate of dead leukemia cells	0.408	days ${}^{-1}$
$\mu_{C}$	Deactivation/decay rate of the chemotherapy drug	11.090	days ${}^{-1}$
$\mu_{CA}$	Deactivation rate of chemotherapy due to killing leukemia cells	10$\mu_{AC}$	$\frac{\mathrm{molecules}}{\mathrm{cells}\times \mathrm{days}}$
$\mu_{CT}$	Deactivation rate of chemotherapy due to killing CAR-T cells	$10\mu_{TC}$	$\frac{\mathrm{molecules}}{\mathrm{cells}\times \mathrm{days}}$
$\varphi_{C}$	Chemotherapy drug (chlorambucil )	$\varphi_{C}\geq 0$	$\frac{\mathrm{molecules}}{\mathrm{days}}$
$\rho_{T}$	Mitosis stimulation/proliferation rate of CAR-T cells	$0<\rho_{T}<\alpha_{T}$	${\left(\mathrm{cells}\times \mathrm{days}\right)}^{-1}$
$\mu_{T}$	Death rate/exhaustion of CAR-T cells	$\frac{1}{14}\leq \mu_{T}\leq \frac{1}{30}$	days ${}^{-1}$
$\mu_{TC}$	Chemotherapy cytotoxicity rate on CAR-T cells	$0<\mu_{TC}<\mu_{AC}$	days ${}^{-1}$
$\varphi_{T}$	CAR-T cell therapy	$\varphi_{T}\geq 0$	$\frac{\mathrm{cells}}{\mathrm{days}}$

**Table 2 pharmaceutics-14-01396-t002:** Chlorambucil concentration and its computed and approximated cytotoxicity rates $\left(\mu_{AC}\right)$ on leukemia cancer cells. Data from columns ‘Concentration in µM’ and ‘Experimental cytotoxicity in h ${}^{-1}$’ were extracted from Appendix A in Guzev et al. [25].

Concentration	Concentration	Experimental	Experimental	Fitted Cytotoxicity
in µM	in mg/L	Cytotoxicity in h ${}^{-1}$	Cytotoxicity in Days${}^{-1}$	$\left[\mu_{\mathrm{AC}}\left(\varkappa \right)\right]\;\mathrm{in}\;{\mathrm{Days}}^{-1}$
0	0	0	0	0
1.5625	0.4753	0.0030	0.0720	0.0651
3.125	0.9507	0.0040	0.0960	0.0974
6.25	1.9013	0.0060	0.1440	0.1447
12.5	3.8026	0.0088	0.2112	0.2128
25	7.6053	0.0126	0.3024	0.3066
50	15.2106	0.0180	0.4320	0.4284

**Table 3 pharmaceutics-14-01396-t003:** Parameters and 95% CI for the chlorambucil cytotoxicity rate (days${}^{-1}$) of Equation (Equation 8) from its concentration in mg/L.

Parameter	Units	Value	95% CI
$\varepsilon_{1}$	L/mg	$4.799\times 10^{-2}$	$\left(4.145\times 10^{-2},5.454\times 10^{-2}\right)$
$\varepsilon_{2}$	Dimensionless	1.693	$\left(1.599,1.786\right)$

**Table 4 pharmaceutics-14-01396-t004:** CAR-T cells cytotoxicity rates (denoted as $\alpha_{T}$ and given in days ${}^{-1}$) estimated by Valle et al. [29] from the Lee et al. [44] phase 1 dose-escalation trial on 21 leukemia patients. Column ‘Fitted cytotoxicity $\left[\alpha_{T}\left(\rho_{T},\mu_{T},\rho_{A}\right)\right]$’ shows the approximated values with Equation (Equation 12).

CAR-T Cells Mitosis	CAR-T Cells Natural	Leukemia Cells	Experimental	Fitted Cytotoxicity
Stimulation Rate $\left[\rho_{T}\right]$	Death Rate $\left[\mu_{T}\right]$	Growth Rate $\left[\rho_{A}\right]$	Cytotoxicity $\left[\alpha_{T}\right]$	$\left[\alpha_{T}\left(\rho_{T},\mu_{T},\rho_{A}\right)\right]$
$2.786\times 10^{-8}$	1/30	1/60	$1.393\times 10^{-7}$	$1.401\times 10^{-7}$
$2.880\times 10^{-8}$	1/30	1/40	$1.440\times 10^{-7}$	$1.441\times 10^{-7}$
$3.170\times 10^{-8}$	1/30	1/20	$1.585\times 10^{-7}$	$1.577\times 10^{-7}$
$3.022\times 10^{-8}$	1/14	1/60	$1.511\times 10^{-7}$	$1.507\times 10^{-7}$
$3.120\times 10^{-8}$	1/14	1/40	$1.560\times 10^{-7}$	$1.554\times 10^{-7}$
$3.392\times 10^{-8}$	1/14	1/20	$1.696\times 10^{-7}$	$1.705\times 10^{-7}$

**Table 5 pharmaceutics-14-01396-t005:** Parameters and 95% CI for the CAR-T cytotoxicity rate Equation (Equation 12).

Parameter	Units	Value	95% CI
$\gamma_{1}$	Dimensionless	$1.256\times 10^{-7}$	$\left(1.219\times 10^{-7},1.294\times 10^{-7}\right)$
$\gamma_{2}$	${\left({\mathrm{days}}^{2}\times \mathrm{cells}\right)}^{-1}$	$1.221\times 10^{-1}$	$\left(9.366\times 10^{-2},1.506\times 10^{-1}\right)$
$\gamma_{3}$	${\left({\mathrm{days}}^{2}\times \mathrm{cells}\right)}^{-1}$	$6.777\times 10^{-2}$	$\left(5.628\times 10^{-2},7.925\times 10^{-2}\right)$

**Table 6 pharmaceutics-14-01396-t006:** Chlorambucil characteristics for the chemoimmunotherapy protocol cases.

Dose	Molecules	Concentration	Cytotoxicity	Cytotoxicity
$C_{\mathrm{mg}}$	$C_{\mathrm{mg}}\times \varsigma$	ϰ	$\mu_{\mathrm{AC}}\left(\varkappa \right)$	$\mu_{\mathrm{TC}}$
40 mg	$7.91834740\times 10^{19}$	8.0 mg/L	0.3145 days ${}^{-1}$	0.0170 days ${}^{-1}$
48 mg	$9.50201688\times 10^{19}$	9.6 mg/L	0.3444 days ${}^{-1}$	0.0186 days ${}^{-1}$
56 mg	$1.10856864\times 10^{20}$	11.2 mg/L	$0.3712\;{\mathrm{days}}^{-1}$	0.0200 days ${}^{-1}$
64 mg	$1.26693558\times 10^{20}$	12.8 mg/L	0.3955 days ${}^{-1}$	0.0214 days ${}^{-1}$

## Data Availability

Data are contained within the article and the Appendix A.

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
