# Peer review of "Chemoimmunotherapy Administration Protocol Design for the Treatment of Leukemia through Mathematical Modeling and In Silico Experimentation"

_pharmaceutics, 2022, doi:10.3390/pharmaceutics14071396_

Round 1

Reviewer 1 Report

This article presented modeled the response of leukemia to chemoimmunotherapy as a nonlinear dynamical system, and explored the treatment protocol design via analytical analysis and numerical simulation. Overall, this is a very good study and is of great interest in the field of mathematical biology/oncology. However, I do not think Pharmaceutics is the best target journal for this manuscript. Most importantly, the presentation of the current manuscript is more like a mathematical paper, which is not matching the structure suggested by the journal. I suggest the authors consider resubmission to another mathematical journal or computational oncology journal. But if the authors do want to revise the current manuscript for submission to Pharmaceutics, several specific suggestions are given below.

1.     Please describe the estimation of cytotoxicity of chemo- and CAR-T cell therapies after the introduction of the modeling system. That is, move lines 108 – 140 behind the end of current section 3 “Mathematical model: leukemia and chemoimmunotherapy”. Without the introduction of equations (6-9) and the definition of parameters, the estimation of a few parameters won't make sense. Also, I think lines 108 – 140 contain too many details for the main manuscript. Many of the calculations can be moved to supplemental materials. Moreover, the pseudocode in Table 5 does not really provide additional information beyond the text so can be eliminated.  

2.     Many definitions in the Table 6 are not clear. For example, both mu_AC and mu_TC are defined as “Cytotoxicity rate due to the chemotherapy drug”. The difference between these two parameters needs to be clearly clarified. Similar comments for other parameters.

3.     The whole section Results should be moved to the “Methods and Materials” section. Analytical derivations are usually not considered a result in the biomedical field.

4.     The numerical simulation analysis in the current Discussion section should be included in the Results section.

5.     There are two Table 1.

6.     Majority of the current Conclusions section should be included in the Discussion. Also, the Discussion section should compare this study with previous related studies, clarify the novelty of this study as compared to previous ones, discuss the potential application and impact of the presented approach in healthcare, and acknowledge the limitations of the present study.

7.     Conclusion section should be a short paragraph summarizing the most important take-home message of this study. 

Author Response

See PDF document.

Reviewer 2 Report

Authors Valle et al. present an interesting pharmacokinetic in silico and mathematical study that successfully answers the question of establishing quantitative conditions for the eradication of CLL cells under a combined chemo and CAR-T cell therapeutic protocol. These conditions are described by the set of inequalities in line 399 of the manuscript, and were established by a thorough analysis of the ODE system in equations (6) to (9). This analysis combines a formal mathematical analysis of the equations, establishing the conditions for physical feasibility of asymptotic cell counts and concentration behaviours, and is reinforced by numerical simulations that further extend the predictions to time varying therapeutic scenarios.

I can not find any scientific objection or flaw in the author’s approach, as I believe that the methods chosen are suited to answer the fundamental questions set forward in the introduction. The authors also go far in justifying the parameters underlying the mathematical representation, making their model quite reliable and the predictions sound. Furthermore, the value of this study is very high since, as far as I know, this kind of mathematical/in silico approach applied to combined therapies has not been attempted before for leukaemia. The prospect of using the findings of this study to improve the therapeutic protocols is very interesting.

 From the presentation point of view, I believe that this submission needs some review as, in general, the work is presented with an excessive level of detail, while, in some parts, the opposite happens, and the authors overlook some needed aspects. Therefore, I would like to encourage the authors to submit a revised version where the following (minor) points should be addressed:

-        The way that the authors refer to tables 1 and 3 is confusing and puzzling: the authors consistently refer to the different columns in these tables by number “Column three…” and yet no explicit numbering is actually shown. Some improvement on the column headers could avoid these implicit numeric references altogether. For example, the column on the right of table 1, “column number 5”, can have a title that reads “cytotoxicity rate predicted from eq (1)”. The column titles are all that is needed to locate the different elements of information in these two tables, and, therefore, there is no need to use “column n” references in the text. I also would like to suggest that table 1 is replaced by a figure with the data from Guzev represented by points and the fitted values from eq (1) by a line, (perhaps using a log scale in y). This would, in my opinion, be a much lighter and elegant way to present the (important) parametrization of eq (1).

-        In table 3, I strongly suggest that the Parameters column is unfolded into 3 columns, rhoT, miuT and rhoA. There is no need for the excessive mathematical style of large braces with enumeration of the 3 parameters.

-        Table 5 should be removed from the text and substituted by the actual listing of the python code in suppl. materials. Table 5 presents several problems. 1- As pseudo-code it is not sufficiently abstract and clear as it uses too much of the specifics of python language. On the other hand, it is not legal python code because of the capitalization of some of the keywords. There is no need for this chimeric pseudo-python code here. The real python code could be presented in the suppl. materials. I think that, for what the authors are trying to present, the real python code is quite readable as it is, as long as you know a little about the capabilities of the most popular python scientific modules. The real important information that the authors are conveying here is that eq. (5) was fitted to data using function curve_fit of module scipy.optimize. That is all there should be indicated in the Methods section of the paper, not this listing. Having said that, the version of scipy must be indicated, as it is even more important than the python version and far more important than the jupyter platform version. Also, important are the “initial guesses” fed to “curve_fit” which must be indicated. If curve_fit() was also used to fit eq (1), authors should state that.

-        Still on the Methods section, and in contrast to the fitting of non-linear equations to data, explicit mentions to the software or modules and corresponding versions used in the numerical solutions of the systems of ODEs are absent from the manuscript.

-        I understand that it is not (yet) mandatory in MDPI journals, but the authors should consider the disclosure of the source code for the analysis via a public git repository hosted in one of the popular services available today for such scientific computing software. That would totally replace the need to list pseudo or actual code in the manuscript.

-        As a very minor point, some computed parameters are not dimensionless and units should be indicated. As a first example, parameter epsilon 1 in equation (1).

-        Some care should be taken to adjust the positioning inside the graphical legends in some figures: the Ad^* is closer to the dash line than it is to the symbol x, which is associated with this value. The same happens with T^* and the same symbol. The authors should fix it and, perhaps, using a legend where the different time-series are indicated in different lines within the legend.

Author Response

See PDF document.

Reviewer 3 Report

The paper reports a novel technique to explore cancer growth under the combined application of CAR-T cells and chlorambucil, considering it as an ODE-described nonlinear dynamical system with the following numerical and nonlinear analysis. The manuscript is well-written, and the underlying idea is clear. The key conclusions of the study are reliable and correspond well with in silico experimental evaluations. I can recommend this paper for publication in MDPI Pharmaceuticals after only minor revisions. Please, find my recommendations below.

1. The introductory part is very long. Many preliminary definitions were previously published, and I believe, the introductory sections can be reduced by proper referencing and citations. Moreover, the conditions and parameters for each numerical experiment must be clearly stated: this increases the reproducibility of the study.

2. It is known that numerical methods can significantly affect the behavior of discrete models of nonlinear systems during the simulation, especially in the case of highly oscillatory and chaotic systems. Which numerical methods were chosen for your in silico experiments? How were the simulation results verified? What was the global truncation error? Which stepsize was taken? How does the stepsize correspond with the overall stability of the finite-difference scheme? All these questions are to be clarified during the revisions.

3. Can the adaptive symmetry approach be used to adjust the stability of the discrete system to better cope with the real data? Please, consider this approach in the discussions section.

4. Is it possible to apply some structural identification techniques based on hybrid synchronization between real data and simulation to your model for its refinement? I believe, such an approach can increase the correspondence between real data and simulation and enrich the dynamics of the model.

Nevertheless, my overall impression is very high and I can recommend this comprehensive manuscript for publication.

Round 2

Reviewer 1 Report

The authors addressed most of my comments. However, I still consider the structure of the Results and Discussion sections needs to be modified before publication.

Guiding the design of chemoinmunotherapy protocol with mathematical modeling is indeed of great interest, and I agree the topic is within the scope of the journal. However, even for the same research project, when targeting different audiences, the focus and structure of the presentation shall be different. I think the current structure of Results and Discussion sections will be very difficult for the audience of Pharmaceutics to follow.

Thus, I still suggest the authors make major modifications to the Methods, Results and Discussion sections. That is, move the current Results section to the Methods, and keep it concise as possible. Move the Discussion section to the Results. And include the comparison to previous studies in the Discussion -- for example, the authors mentioned in the reply that other in silico works have shortcomings in comparinson to the present study. This need to be explicitly discussed in the manuscript, with the referred studies cited. 

Author Response

Reviewer 1:

We greatly appreciate all comments on our revised manuscript and the time invested in our work. Hopefully, the following responses to each of your concerns may be suffcient to move forward in our submission.

Following your suggestions regarding the organization of our paper, our changes are as follows:

  1. Section 2 was previously Section 3 but was moved to Section 2 to follow the new organization of the paper.
  2. Materials and methods were updated to better present our results for the main audience of the Journal as suggested by Reviewer 1. Subsections are now as follows and we tried to keep it as concise as possible: 3.1. Estimating cytotoxicity of chemotherapy and CAR-T cells to cancer cells; 3.2. Localization of compact invariant sets (here some steps of the mathematical analysis were reduced); 3.3. Global asymptotic stability: Leukemia cells eradication; 3.4. Persistence: Leukemia and CAR-T cells.
  3. The discussion section of version 1 and 2 is now our Results section as suggested, i.e., the numerical simulations and protocol design.
  4. The discussion section deals now only with implications of mathematical and numerical simulation results.

Round 3

Reviewer 1 Report

The authors have addressed my concerns and suggestions in the revised version.